# ByPE-VAE: Bayesian Pseudocoresets Exemplar VAE

**Qingzhong Ai**[1]     **Lirong He**[1]     **Shiyu Liu**[1]     **Zenglin Xu**[2,3,*]

[1]School of Computer Science and Engineering,
University of Electronic Science and Technology of China, Chengdu China
[2]School of Science and Technology,
Harbin Institute of Technology Shenzhen, Shenzhen China
[3]Department of Network Intelligence,
Peng Cheng National Lab, Shenzhen, China
{qzai,lirong_he}@std.uestc.edu.cn, shyu.liu@foxmail.com, xuzenglin@hit.edu.cn

## Abstract

Recent studies show that advanced priors play a major role in deep generative models. Exemplar VAE, as a variant of VAE with an exemplar-based prior, has achieved impressive results. However, due to the nature of model design, an exemplar-based model usually requires vast amounts of data to participate in training, which leads to huge computational complexity. To address this issue, we propose Bayesian Pseudocoresets Exemplar VAE (ByPE-VAE), a new variant of VAE with a prior based on Bayesian pseudocoreset. The proposed prior is conditioned on a small-scale pseudocoreset rather than the whole dataset for reducing the computational cost and avoiding overfitting. Simultaneously, we obtain the optimal pseudocoreset via a stochastic optimization algorithm during VAE training aiming to minimize the Kullback-Leibler divergence between the prior based on the pseudocoreset and that based on the whole dataset. Experimental results show that ByPE-VAE can achieve competitive improvements over the state-of-the-art VAEs in the tasks of density estimation, representation learning, and generative data augmentation. Particularly, on a basic VAE architecture, ByPE-VAE is up to 3 times faster than Exemplar VAE while almost holding the performance. Code is available at `https://github.com/Aiqz/ByPE-VAE`.

## 1  Introduction

Deep generative models that learn implicit data distribution from the enormous amount of data have received widespread attention to generating highly realistic new samples in machine learning. In particular, due to the utilization of the reparameterization trick and variational inference for optimization, Variational Autoencoders (VAEs) [1, 2] stand out and have demonstrated significant successes for dimension reduction [3], learning representations [4], and generating data [5]. In addition, various variants of VAE have been proposed conditioned on advanced variational posterior [6, 7, 8] or powerful decoders [9, 10].

It is worth noting that the prior in the typical VAE is a simple standard normal distribution that is convenient to compute while ignores the nature of the data itself. Moreover, a large number of experiments have empirically demonstrated that simplistic priors could produce the phenomena of over-regularization and posterior collapse, and finally cause poor performance [5, 11]. Hence, many researchers have worked to develop more complex priors to enhance the capacity of the variational posterior. In this line, Tomczak *et al.* [12] introduces a more flexible prior named VampPrior, which is a mixture of variational posteriors based on pseudo-inputs to alleviate the problems like overfitting

---

*Corresponding Author.

35th Conference on Neural Information Processing Systems (NeurIPS 2021).

and high computational cost. However, the way in which the pseudo-inputs are obtained is not interpretable. Recently, Norouzi *et al.* [13] develops an Exemplar VAE with a non-parametric prior based on an exemplar-based method, achieving excellent performance. To ensure the performance and the generation diversity, the exemplar set needs to be large enough, and usually the entire training data is utilized. Obviously, this leads to huge computational complexity. Even though Exemplar VAE further employs approximate nearest neighbor search to speed up the training process, the number of nearest neighbors should be as large as possible to ensure performance. In a nutshell, Exemplar VAE is computationally expensive.

To address such issues, we develop a new prior for VAE that is inspired by the paradigm of coresets [14]. The coreset is a powerful tool aiming to find a small weighted subset for efficiently approximating the entire original dataset. Therefore, rather than using the large-scale training data directly, we seek to design a prior conditioned on a coreset, which greatly reduces the computational complexity and prevents overfitting. In practice, to better incorporate this idea with the framework of VAE, we further employ a specific form of coresets, namely Bayesian pseudocoresets [15], which is known as a small weighted subset of the pseudodata points, resulting in a Bayesian pseudocoreset based prior. With this prior, we gain a new variant of VAE called Bayesian Pseudocoresets Exemplar VAE (ByPE-VAE). To sample from the ByPE-VAE, we first take a pseudodata point from the pseudocoreset according to its weight, and then transform it into a latent representation using the learned prior. Then a decoder is used to transform the latent representation into a new sample.

A crucial part of ByPE-VAE is to obtain the optimal pseudocoreset. We formulate this process as a variational inference problem where the pseudodata points and corresponding weights are the parameters of variational posterior approximation. More precisely, to seek the optimal pseudocoreset, we minimize the Kullback-Leibler (KL) divergence between the prior based on the pseudocoreset and that based on the entire dataset. This processing ensures that the learned prior is actually an approximation of the prior conditioned on whole training data. Thus, it is fundamentally different from general pseudodata based priors, like VampPrior. For optimization, we adopt a two-step alternative search strategy to learn two types of parameters of ByPE-VAE, which refer to the parameters in the VAE framework and the pseudodata points and corresponding weights in the pseudocoreset. In particular, we iteratively optimize one of the two types of parameters while keeping the other one fixed until convergence.

Finally, we compare ByPE-VAE with several state-of-the-art VAEs in a number of tasks, including density estimation, representation learning and generative data augmentation. Experimental results demonstrate the effectiveness of ByPE-VAE on Dynamic MNIST, Fashion MNIST, CIFAR10, and CelebA. Additionally, to validate the efficiency of our model, we measure the running time on a basic VAE architecture. Compared to the Exemplar VAE, ByPE-VAE is up to 3 times speed-up without losing performance on Dynamic MNIST, Fashion MNIST, and CIFAR10.

## 2 Preliminaries

Before presenting the proposed model, we first introduce some preliminaries, namely Exemplar VAE and Bayesian pseudocoresets. Throughout this paper, vectors are denoted by bold lowercase letters, whose subscripts indicate their order, and matrices are denoted by upper-case letters.

### 2.1 Exemplar VAE

Exemplar VAE is regarded as a variant of VAE that integrates the exemplar-based method into VAE for the sake of seeking impressive image generation. Specifically, it first draws a random exemplar $\mathbf{x}_n$ using uniform distribution from the training data $X = \{\mathbf{x}_n\}_{n=1}^N$ (where $N$ denotes the sample amount), then transforms an exemplar $\mathbf{x}_n$ into latent variable $\mathbf{z}$ using an example-based prior $r_\phi(\mathbf{z} \mid \mathbf{x}_n)$, finally generates observable data $\mathbf{x}$ by a decoder $p_\phi(\mathbf{x} \mid \mathbf{z})$. The parametric transition distribution $T_{\phi,\theta}(\mathbf{x} \mid \mathbf{x}_n)$ of exemplar-based generative models can be expressed as

$$T_{\phi,\theta}(\mathbf{x} \mid \mathbf{x}_n) = \int_{\mathbf{z}} r_\phi(\mathbf{z} \mid \mathbf{x}_n) p_\theta(\mathbf{x} \mid \mathbf{z}) d\mathbf{z}, \tag{1}$$

where $\phi$ and $\theta$ denote corresponding parameters. Assuming that $\mathbf{x}$ is independent to $\mathbf{x}_n$ conditioned on the latent variable $\mathbf{z}$ and marginalizing over the latent variable $\mathbf{z}$, the objective $O(\theta, \phi; \mathbf{x}, X)$ of

Exemplar VAE can be formulated as

$$\log p(\mathbf{x}; X, \theta, \phi) = \log \sum_{n=1}^{N} \frac{1}{N} T_{\phi,\theta}(\mathbf{x}|\mathbf{x}_n) \geq \mathbb{E}_{q_\phi(\mathbf{z}|\mathbf{x})} \log p_\theta(\mathbf{x}|\mathbf{z}) - \mathbb{E}_{q_\phi(\mathbf{z}|\mathbf{x})} \log \frac{q_\phi(\mathbf{z}|\mathbf{x})}{\sum_{n=1}^{N} r_\phi(\mathbf{z}|\mathbf{x}_n)/N}$$
$$= O(\theta, \phi; \mathbf{x}, X), \tag{2}$$

where $O(\theta, \phi; \mathbf{x}, X)$ is known as the evidence lower bound (ELBO). From the Eq. (2), it can be derived that the difference between Exemplar VAE and typical VAE is the definition of the prior $p(\mathbf{z})$ in the second term. The prior of Exemplar VAE is defined as a mixture form, that is $p(\mathbf{z} \mid X) = \sum_{n=1}^{N} r_\phi(\mathbf{z} \mid \mathbf{x}_n)/N$. The variational posterior $q_\phi(\mathbf{z} \mid \mathbf{x})$ and the exemplar-based prior $r_\phi(\mathbf{z} \mid \mathbf{x}_n)$ are assumed to be Gaussian distributions whose parameters are fulfilled by neural networks. Note that, the computational cost of this training process is related to the number of exemplars which usually set to be the entire training data. This indicates that Exemplar VAE is computationally expensive since the amount of training data is generally huge.

## 2.2 Bayesian Pseudocoresets

Bayesian pseudocoresets is a method of coreset construction based on variational inference, which constructs a weighted set of synthetic "pseudodata" instead of the original dataset during inference. First, the goal of this method is to approximate expectations under the posterior $\pi(\psi)$ with the parameter $\psi$, which is formulated as

$$\pi(\psi) = \frac{1}{Z} \exp\left(\sum_{n=1}^{N} f(\mathbf{x}_n, \psi)\right) \pi_0(\psi), \tag{3}$$

where $f(\cdot, \psi)$ denotes a potential function, and usually is a log-likelihood function. $\pi_0(\psi)$ is the prior, and $Z$ is the normalization constant. Instead of directly inferring the posterior $\pi(\psi)$, the Bayesian pseudocoreset employs a weighted set of pseudodata points to approximate the true posterior $\pi(\psi)$, which is given by

$$\pi_{U,\mathbf{w}}(\psi) = \frac{1}{\tilde{Z}_{U,\mathbf{w}}} \exp\left(\sum_{m=1}^{M} w_m f(\mathbf{u}_m, \psi)\right) \pi_0(\psi), \tag{4}$$

where $U = \{\mathbf{u}_m\}_{m=1}^{M}$ represents $M$ pseudodata points $\mathbf{u}_m \in \mathbb{R}^d$, $\mathbf{w} = \{w_m\}_{m=1}^{M}$ denotes non-negative weights, and $\tilde{Z}_{U,\mathbf{w}}$ is the corresponding normalization constant. Finally, this model obtains the optimal pseudodata points and their weights by minimizing the KL divergence, as follows,

$$U^\star, \mathbf{w}^\star = \underset{U,\mathbf{w}}{\operatorname{argmin}} \, D_{\mathrm{KL}}\left(\pi_{U,\mathbf{w}} \| \pi\right). \tag{5}$$

This formulation can reduce the computational cost by decreasing data redundancy.

## 3 Bayesian Pseudocoresets Exemplar VAE

To generate a new observation $\mathbf{x}$, the Exemplar VAE requires a large collection of exemplars from $X = \{\mathbf{x}_n\}_{n=1}^{N}$ to guide the whole process, as shown in Eq. (2). One can see that the greater the number of exemplars set, the richer the prior information can be obtained. In practice, to ensure performance, the number of exemplars is relatively large which is generally set to the size of the entire training data. This leads to huge computational costs in the training process of the Exemplar VAE. To overcome such a issue, inspired by the paradigm of Bayesian pseudocoresets, we adopt $M$ pseudodata points $U = \{\mathbf{u}_m\}_{m=1}^{M}$ with corresponding weights $\mathbf{w} = \{w_m\}_{m=1}^{M}$ to denote exemplars, importantly $M \ll N$. That is, the original exemplars $X$ are approximated by a small weighted set of pseudodata points known as a Bayesian pseudocoreset. The framework can be expressed as

$$\log p(\mathbf{x} \mid X, \theta) = \log \sum_{n=1}^{N} \frac{1}{N} T_\theta(\mathbf{x} \mid \mathbf{x}_n) \approx \log \sum_{m=1}^{M} \frac{w_m}{N} T_\theta(\mathbf{x} \mid \mathbf{u}_m) = \log p(\mathbf{x} \mid U, \mathbf{w}, \theta), \quad (6)$$

where $w_m \geq 0$ $(m = 1, \cdots, M)$ and $\|\mathbf{w}\|_1 = N$. Further, we integrate this approximated framework with VAE by introducing a latent variable $\mathbf{z}$. Then the parametric function is given by

$$T_{\phi,\theta}(\mathbf{x} \mid \mathbf{u}_m) = \int_{\mathbf{z}} r_\phi(\mathbf{z} \mid \mathbf{u}_m) p_\theta(\mathbf{x} \mid \mathbf{z}) d\mathbf{z}, \tag{7}$$

where $r_\phi(\mathbf{z} \mid \mathbf{u}_m)$ with parameter $\phi$ denotes a pseudodata based prior for generating $\mathbf{z}$ from a pseudodata point $\mathbf{u}_m$. $p_\theta(\mathbf{x} \mid \mathbf{z})$ with parameter $\theta$ represents the decoder for generating the observation $\mathbf{x}$ from $\mathbf{z}$. Similarly, we assume that an observation $\mathbf{x}$ is independent from a pseudodata point $\mathbf{u}_m$ conditional on $\mathbf{z}$ to simplify the formulation and optimization.

In general, we desire to maximize the marginal log-likelihood $\log p(\mathbf{x})$ for learning, however, this is intractable since we have no ability to integrate the complex posterior distributions out. Now, we focus on maximizing the evidence lower bound (ELBO) derived by Jensen's inequality, as follows,

$$\log p(\mathbf{x}; U, \mathbf{w}, \theta, \phi) = \log \sum_{m=1}^{M} \frac{w_m}{N} T_{\phi,\theta}(\mathbf{x} \mid \mathbf{u}_m) = \log \sum_{m=1}^{M} \frac{w_m}{N} \int_{\mathbf{z}} r_\phi(\mathbf{z} \mid \mathbf{u}_m) p_\theta(\mathbf{x} \mid \mathbf{z}) d\mathbf{z} \quad (8)$$

$$\geq \mathbb{E}_{q_\phi(\mathbf{z}|\mathbf{x})} \log p_\theta(\mathbf{x} \mid \mathbf{z}) - \mathbb{E}_{q_\phi(\mathbf{z}|\mathbf{x})} \log \frac{q_\phi(\mathbf{z} \mid \mathbf{x})}{\sum_{m=1}^{M} w_m r_\phi(\mathbf{z} \mid \mathbf{u}_m)/N}$$

$$\equiv O(\theta, \phi, U, \mathbf{w}; \mathbf{x}), \tag{9}$$

where $q_\phi(\mathbf{z} \mid \mathbf{x})$ represents the approximate posterior distribution. And $O(\theta, \phi, U, \mathbf{w}; \mathbf{x})$ is defined as the objective function of the ByPE-VAE to optimize parameters $\theta$ and $\phi$. The specific derivation can be found in Supp.A. As we can see from Eq. (9), the difference between the ByPE-VAE and other variants of VAE is the formulation of the prior $p(\mathbf{z})$ in the second term. In detail, the prior of the ByPE-VAE is a weighted mixture model prior, in which each component is conditioned on a pseudodata point and the corresponding weight, i.e., $p(\mathbf{z}|U, \mathbf{w}) = \sum_{m=1}^{M} w_m r_\phi(\mathbf{z}|\mathbf{u}_m)/N$. In contrast, the prior of the Exemplar VAE is denoted as $p(\mathbf{z}|X) = \sum_{n=1}^{N} r_\phi(\mathbf{z}|\mathbf{x}_n)/N$.

As shown in Eq. (9), the ByPE-VAE includes two encoder networks, namely $q_\phi(\mathbf{z} \mid \mathbf{x})$ and $r_\phi(\mathbf{z} \mid \mathbf{u}_m)$. And the distributions of $q_\phi(\mathbf{z} \mid \mathbf{x})$ and $r_\phi(\mathbf{z} \mid \mathbf{u}_m)$ are both designed as Gaussian distributions. According to the analysis in [16], the optimal prior is the form of aggregate posterior. Inspired by this report, we make the prior be coupled with the variational posterior, as follows,

$$q_\phi(\mathbf{z} \mid \mathbf{x}) = \mathcal{N}(\mathbf{z} \mid \boldsymbol{\mu}_\phi(\mathbf{x}), \Lambda_\phi(\mathbf{x})), \tag{10}$$

$$r_\phi(\mathbf{z} \mid \mathbf{u}_m) = \mathcal{N}(\mathbf{z} \mid \boldsymbol{\mu}_\phi(\mathbf{u}_m), \sigma^2 I). \tag{11}$$

We employ the same parametric mean function $\mu_\phi$ for two encoder networks for better incorporation. However, the covariance functions of the two encoder networks are different. Specifically, the variational posterior uses a diagonal covariance matrix function $\Lambda_\phi$, while each component of the mixture model prior uses an isotropic Gaussian with a scalar parameter $\sigma^2$. Note that the scalar parameter $\sigma^2$ is shared by each component of the mixture model prior for effective computation. Then, we can express the log of the weighted pseudocoreset based prior $\log p_\phi(\mathbf{z} \mid U, \mathbf{w})$ as

$$\log p_\phi(\mathbf{z} \mid U, \mathbf{w}) = -d_\mathbf{z} \log(\sqrt{2\pi}\sigma) - \log N + \log \sum_{m=1}^{M} w_m \exp \frac{-\|\mathbf{z} - \boldsymbol{\mu}_\phi(\mathbf{u}_m)\|^2}{2\sigma^2}, \tag{12}$$

where $d_\mathbf{z}$ denotes the dimension of $\mathbf{z}$. Based on the formulation of Eq. (12), we can further obtain the objective function of the ByPE-VAE, as follows,

$$O(\theta, \phi, U, \mathbf{w}; X) = \mathbb{E}_{q_\phi(\mathbf{z}|\mathbf{x})} \left[ \log \frac{p_\theta(\mathbf{x} \mid \mathbf{z})}{q_\phi(\mathbf{z} \mid \mathbf{x})} + \log \sum_{m=1}^{M} \frac{w_m}{(\sqrt{2\pi}\sigma)^{d_\mathbf{z}}} \exp \frac{-\|\mathbf{z} - \boldsymbol{\mu}_\phi(\mathbf{u}_m)\|^2}{2\sigma^2} \right], \tag{13}$$

$$\propto \sum_{i=1}^{N} \mathbb{E}_{q_\phi(\mathbf{z}|\mathbf{x}_i)} \left[ \log \frac{p_\theta(\mathbf{x}_i \mid \mathbf{z})}{q_\phi(\mathbf{z} \mid \mathbf{x}_i)} + \log \sum_{m=1}^{M} \frac{w_m}{(\sqrt{2\pi}\sigma)^{d_\mathbf{z}}} \exp \frac{-\|\mathbf{z} - \boldsymbol{\mu}_\phi(\mathbf{u}_m)\|^2}{2\sigma^2} \right], \tag{14}$$

where the constant $-\log N$ is omitted for convenience. For $\mathbb{E}_{q_\phi(\mathbf{z}|\mathbf{x}_i)}$, we employ the reparametrization trick to generate samples. Note that, for the standard VAE with a Gaussian prior, the process of generating new observations involves only the decoder network after training. However, to generate a new observation from ByPE-VAE, we not only require the decoder network, but also the learned Bayesian pseudocoreset and a pseudodata point based prior $r_\phi$. We summarize the generative process of the ByPE-VAE in Algorithm 1.

**Algorithm 1:** The Generative Process of ByPE-VAE

---

**Input** : Pseudocoreset $\{U, \mathbf{w}\}$, Decoder $p_\theta$, Prior $r_\phi$.
**Output:** A generated observation $\mathbf{x}$.
**Step.1** Sample $\mathbf{u}_m \sim \text{Multi}(\cdot \mid U, \frac{\mathbf{w}}{N})$ for obtaining a pseudodata point $\mathbf{u}_m$ from the learned pseudocoreset.
**Step.2** Sample $\mathbf{z} \sim r_\phi(\cdot \mid \mathbf{u}_m)$ using the pseudodata point based prior $r_\phi$ for obtaining the latent representation $\mathbf{z}$.
**Step.3** Sample $\mathbf{x} \sim p_\theta(\cdot \mid \mathbf{z})$ using the decoder $p_\theta$ for generating a new observation $\mathbf{x}$.

---

However, more importantly, the whole process above holds only if the pseudocoreset is an approximation of all exemplars or training data. To ensure this, we first re-represent frameworks of $p_\phi(\mathbf{z} \mid X)$ and $p_\phi(\mathbf{z} \mid U, \mathbf{w})$ from the Bayesian perspective. Concretely, $p_\phi(\mathbf{z} \mid X)$ and $p_\phi(\mathbf{z} \mid U, \mathbf{w})$ are also viewed as posteriors conditioned on the likelihood function $p_\theta(\cdot \mid \mathbf{z})$ and a certain prior $p_0(\mathbf{z})$, as follows,

$$p_\phi(\mathbf{z} \mid X) = \frac{1}{Z} \exp\left(\sum_{n=1}^{N} \log p_\theta(\mathbf{x}_n \mid \mathbf{z})\right) p_0(\mathbf{z}), \tag{15}$$

$$p_\phi(\mathbf{z} \mid U, \mathbf{w}) = \frac{1}{\tilde{Z}_{U,\mathbf{w}}} \exp\left(\sum_{m=1}^{M} w_m \log p_\theta(\mathbf{u}_m \mid \mathbf{z})\right) p_0(\mathbf{z}), \tag{16}$$

where $Z$ and $\tilde{Z}_{U,\mathbf{w}}$ are their respective normalization constants, $p_\theta(\cdot \mid \mathbf{z})$ here specifically refers to the decoder. Then, we develop this problem into a variational inference problem, where the pseudodata points and the corresponding weights are the parameters of the variational posterior approximation followed [15]. Specifically, we construct the pseudocoreset by minimizing the KL divergence in terms of the pseudodata points and the weights, as follows,

$$U^\star, \mathbf{w}^\star = \underset{U,\mathbf{w}}{\arg\min} \, \mathrm{D}_{\mathrm{KL}}\left(p_\phi(\mathbf{z} \mid U, \mathbf{w}) \| p_\phi(\mathbf{z} \mid X)\right). \tag{17}$$

Further, the gradients of $\mathrm{D}_{\mathrm{KL}}$ in Eq. (17) with respect to the pseudodata point $\mathbf{u}_m$ and the weights $\mathbf{w}$ are given by

$$\nabla_{\mathbf{u}_m} \mathrm{D}_{\mathrm{KL}} = -w_m \, \mathrm{Cov}_{U,\mathbf{w}}\left[\nabla_U \log p_\theta(\mathbf{u}_m \mid \mathbf{z}), \log p_\theta(X \mid \mathbf{z})^T \mathbf{1}_N - \log p_\theta(U \mid \mathbf{z})^T \mathbf{w}\right], \tag{18}$$

$$\nabla_{\mathbf{w}} \mathrm{D}_{\mathrm{KL}} = -\, \mathrm{Cov}_{U,\mathbf{w}}\left[\log p_\theta(U \mid \mathbf{z}), \log p_\theta(X \mid \mathbf{z})^T \mathbf{1}_N - \log p_\theta(U \mid \mathbf{z})^T \mathbf{w}\right], \tag{19}$$

where $\mathrm{Cov}_{U,\mathbf{w}}$ denotes the covariance operator for the $p_\phi(\mathbf{z} \mid U, \mathbf{w})$, and $\mathbf{1}_N \in \mathbb{R}^N$ represents the vector of all 1 entries. In practice, we adopt a black-box stochastic algorithm to obtain the optimal pseudodata points and weights. Details of these derivations are provided in Supp.B and C.

Note that ByPE-VAE involves two types of parameters, namely the parameters $\theta$ and $\phi$ in the VAE framework and the parameters $U$ and $\mathbf{w}$ in the pseudocoreset. For optimization, we use a two-step alternative optimization strategy. In detail, (i) update $\theta$ and $\phi$ with fixed pseudocoreset $\{U, \mathbf{w}\}$, and (ii) update $U$ and $\mathbf{w}$ with fixed $\theta$ and $\phi$. Steps (i) and (ii) are iteratively implemented until convergence. The detailed optimization algorithm is shown in Algorithm 2. One can see that the computation does not scale with $N$, but rather with the number of pseudocoreset points $M$, which greatly reduces the computational complexity and also prevents overfitting. Also note that, rather than be updated every epoch, the pseudocoreset $\{U, \mathbf{w}\}$ is updated by every $k$ epochs. And $k$ is set to 10 in the experiments.

## 4 Related Works

Variational Autoencoders (VAEs) [1, 2] are effectively deep generative models that utilize the variational inference and reparameterization trick for dimension reduction [3], learning representations [4], knowledge base completion[17], and generating data [5]. And various variants of VAE have been proposed conditioned on advanced variational posterior [6, 7, 8], powerful decoders [9, 10] and flexible priors [10, 18, 12, 13]. As for the prior, the standard VAE takes the normal distribution as the prior, which may lead to the phenomena of over-regularization and posterior collapse and further affect the performance for density estimation [5, 11]. In the early stage, VAEs apply more complex

---

**Algorithm 2:** The Optimization Algorithm for ByPE-VAE

---

**Input** : Training data $X \equiv \{\mathbf{x}_n\}_{n=1}^N$, batch size $B$, training epochs $T$, learning rate $\gamma_t$, sample size $S$,
pseudocoreset size $M$, update interval $k$
Decoder $p_\theta$, variational posterior $q_\phi$, weighted pseudocoreset based prior $p_\phi$
Initialized pseudocoreset by which $\mathcal{B} \sim$ UnifSubset $([N], M), \mathcal{B} := \{b_1, \quad \ldots, b_M\}$
$\mathbf{u}_m \leftarrow \mathbf{x}_{b_m}, \quad w_m \leftarrow N/M, \quad m = 1, \cdots, M$
**Output** : Parameters $\theta$ and $\phi$, Pseudocoreset $\{U, \mathbf{w}\}$

---

1 **for** $t = 1, \cdots, T$ **do**
    /\* Optimize VAE parameters $\theta$ and $\phi$                                        \*/
2     $\mathbf{w} \leftarrow \mathbf{w} + (N/M - \mathbf{w}.mean)$    *Centralized to* $\mathbf{w}$
3     Evaluate the ByPE-VAE objective using Eq. (14), and update $\theta$ and $\phi$ using the ADAM
    /\* Optimize pseudocoreset $\mathbf{u}_m$ and $w_m, m = 1, \cdots, M$                \*/
4     **if** $t/k = 0$ **then**
5         Take $S$ samples from current pseudocoreset posterior $p_\phi(\mathbf{z}|U, \mathbf{w})$, namely $(\mathbf{z})_{s=1}^S \sim p_\phi(\mathbf{z}|U, \mathbf{w})$
6         Obtain a mini-batch of $B$ datapoints $\mathcal{B} \sim$ UnifSubset $([N], B)$
7         **for** $s = 1, ..., S$ **do**
8             $\mathbf{g}_s \leftarrow \left(\log p_\theta(\mathbf{x}_b|\mathbf{z}_s) - 1/S \sum_{s'=1}^S \log p_\theta(\mathbf{x}_b|\mathbf{z}_{s'})\right)_{b \in \mathcal{B}} \in \mathbb{R}^B$
9             $\tilde{\mathbf{g}}_s \leftarrow \left(\log p_\theta(\mathbf{u}_m|\mathbf{z}_s) - 1/S \sum_{s'=1}^S \log p_\theta(\mathbf{u}_m|\mathbf{z}_{s'})\right)_{m=1}^M \in \mathbb{R}^M$
10             **for** $m = 1, ..., M$ **do**
11                 $\tilde{\mathbf{h}}_{m,s} \leftarrow \nabla_U \log p_\theta(\mathbf{u}_m|\mathbf{z}_s) - 1/S \sum_{s'=1}^S \nabla_U \log p_\theta(\mathbf{u}_m|\mathbf{z}_{s'})) \in \mathbb{R}^d$
12         $\hat{\nabla}_{\mathbf{w}} \leftarrow -1/s \sum_{s=1}^S \tilde{\mathbf{g}}_s \left(N/B\mathbf{g}_s^T 1 - \tilde{\mathbf{g}}_s^T \mathbf{w}\right)$
13         **for** $m = 1, ..., M$ **do**
14             $\hat{\nabla}_{\mathbf{u}_m} \leftarrow -w_m^1/S \sum_{s=1}^S \tilde{\mathbf{h}}_{m,s} \left(N/B\mathbf{g}_s^T 1 - \tilde{\mathbf{g}}_s^T \mathbf{w}\right)$
15         $\mathbf{w} \leftarrow \max\left(\mathbf{w} - \gamma_t \hat{\nabla}_{\mathbf{w}}, 0\right)$
16         **for** $m = 1, ..., M$ **do**
17             $\mathbf{u}_m \leftarrow \mathbf{u}_m - \gamma_t \hat{\nabla}_{\mathbf{u}_m}$

---

priors, such as the Dirichlet process prior [19], the Chinese Restaurant Process prior [20], to improve the capacity of the variational posterior. However, these methods can only be trained with specific tricks and learning methods. Chen *et al.* [10] employs the autoregressive prior which is then along with a convolutional encoder and an autoregressive decoder to ensure the performance of generation.

Tomczak *et al.* [12] introduces a variational mixture of posteriors prior (VampPrior) conditioned on a set of pseudo-inputs aiming at approximating the aggregated posterior. The intent of our method is similar to the VampPrior with respect to the use of pseudo-inputs. Nevertheless, there is a fundamental difference. The pseudo-inputs in [12] are regarded as hyperparameters of the prior and are obtained through backpropagation, while the pseudo-inputs of our model are the Bayesian pseudocoreset and are optimized through variational inference. Therefore, the pseudo-inputs learned by our model could approximate all the original data, with the weighting operation carried on. In other words, our model is easier to understand in terms of interpretability.

Exemplar VAE [13] is a variant of VAE with a non-parametric prior based on an exemplar-based method to learn desirable hidden representations. Exemplar VAE takes all training data to its exemplar set instead of pseudo-inputs. This computational cost is expensive since the amount of training data can be huge. Hence, Exemplar VAE further presents the approximate kNN search to reduce the cost. However, this technique could reduce the effectiveness of the algorithm and is used only in the training process. Our model introduces the Bayesian pseudocoreset under the Exemplar VAE framework, which improves not only the computational speed, but also the performance of VAEs.

In addition, a memory-augmented generative model with the discrete latent variable [21] is proposed to improve generative models. Our model can be considered as a VAE with additional memory. There are two essential differences between our model and [21]. First, the pseudo-inputs are based on the Bayesian pseudocoreset which is easy to interpret. Second, our model doesn't need a normalized categorical distribution.

# 5 Experiments

**Experimental setup** We evaluate the ByPE-VAE on four datasets across several tasks based on multiple network architectures. Specifically, the tasks involve density estimation, representation learning and data augmentation, the used four datasets include MNIST, Fashion-MNIST, CIFAR10, and CelebA, respectively. Following [13], for the first three datasets, we conduct experiments on three different VAE architectures, namely a VAE based on MLP with two hidden layers, an HVAE based on MLP with two stochastic layers, and a ConvHVAE based on CNN with two stochastic layers. Following [22], we adopt the convolutional architecture for CelebA. In addition, we measure the running time on the first network architecture for three datasets. The ADAM algorithm with normalized gradients [23, 24, 25] is used for optimization and learning rate is set to 5e-4. And we use KL annealing for 100 epochs and early-stopping with a look ahead of 50 epochs. In addition, the weights of the neural networks are initialized according to [26]. Following [12] and [13], we use Importance Weighted Autoencoders (IWAE) [27] with 5000 samples for density estimation.

## 5.1 Density Estimation

To validate the effectiveness of ByPE-VAE, we compare ByPE-VAE with state-of-the-art methods for each architecture, namely a Gaussian prior, a VampPrior, and an Exemplar prior. In order to ensure the fairness of the comparison, the pseudo-inputs size of VampPrior, the exemplars size of Exemplar prior and the pseudocoreset size of ByPE-VAE are set to the same value, which is 500 in all of the experiments except 240 for CelebA. Since the data in CIFAR10 is usually processed as continuous values, we preprocess all used datasets into continuous values in the range of $[0, 1]$ in pursuit of uniformity. This is also beneficial for the pseudocoreset update. We further employ mean square (MSE) error as the reconstruction error. The results are shown in Table 1, from which one can see that ByPE-VAEs outperform other models in all cases.

| Method | Dynamic MNIST | Fashion MNIST | CIFAR10 |
|---|---|---|---|
| VAE w/ Gaussian prior | $24.41 \pm 0.06$ | $21.43 \pm 0.10$ | $72.21 \pm 0.08$ |
| VAE w/ VampPrior | $23.65 \pm 0.03$ | $20.87 \pm 0.01$ | $71.97 \pm 0.05$ |
| VAE w/ Exemplar prior | $23.83 \pm 0.04$ | $21.00 \pm 0.01$ | $72.55 \pm 0.05$ |
| **ByPE-VAE (ours)** | $\mathbf{23.61} \pm 0.03$ | $\mathbf{20.85} \pm 0.01$ | $\mathbf{71.91} \pm 0.02$ |
| HVAE w/ Gaussian prior | $23.82 \pm 0.04$ | $21.04 \pm 0.03$ | $71.63 \pm 0.06$ |
| HVAE w/ VampPrior | $23.54 \pm 0.03$ | $20.83 \pm 0.02$ | $71.54 \pm 0.04$ |
| HVAE w/ Exemplar prior | $23.58 \pm 0.03$ | $20.95 \pm 0.02$ | $71.77 \pm 0.05$ |
| **ByPE-HVAE (ours)** | $\mathbf{23.48} \pm 0.02$ | $\mathbf{20.82} \pm 0.01$ | $\mathbf{71.38} \pm 0.01$ |
| ConvHVAE w/ Gaussian prior | $23.16 \pm 0.05$ | $20.76 \pm 0.01$ | $70.83 \pm 0.05$ |
| ConvH VAE w/ VampPrior | $22.94 \pm 0.02$ | $20.59 \pm 0.01$ | $70.61 \pm 0.06$ |
| ConvHVAE w/ Exemplar prior | $22.92 \pm 0.03$ | $20.62 \pm 0.00$ | $70.83 \pm 0.19$ |
| **ByPE-ConvHVAE (ours)** | $\mathbf{22.84} \pm 0.02$ | $\mathbf{20.58} \pm 0.01$ | $\mathbf{70.55} \pm 0.03$ |

Table 1: Density estimation on Dynamic MNIST, Fashion MNIST, and CIFAR10 based on different network architectures for four methods.

According to the generation process of ByPE-VAE (as shown in Algorithm 1), we generate a set of samples, given in Fig.1. The generated samples in each plate are based on the same pseudodata point. As shown in Fig.1, ByPE-VAE can generate high-quality samples with various identifiable features of the data while inducing a cluster without losing diversity. For the datasets with low diversity, such as MNIST and Fashion MNIST, these samples could retain the content of the data. For more diverse datasets (such as CelebA), although the details of the generated samples are different, they also show clustering effects on certain features, such as background, hairstyle, hair color, and face direction. This phenomenon is probably due to the use of the pseudocoreset during the training phase. In addition, we conduct the interpolation in the latent space on CelebA shown in Fig.2, which implies that the latent space learned by our model is smooth and meaningful.

## 5.2 Representation Learning

Since the structure of latent space also reveals the quality of the generative model, we further report the latent representation of ByPE-VAE. First, we compare ByPE-VAE with VAE with a Gaussian

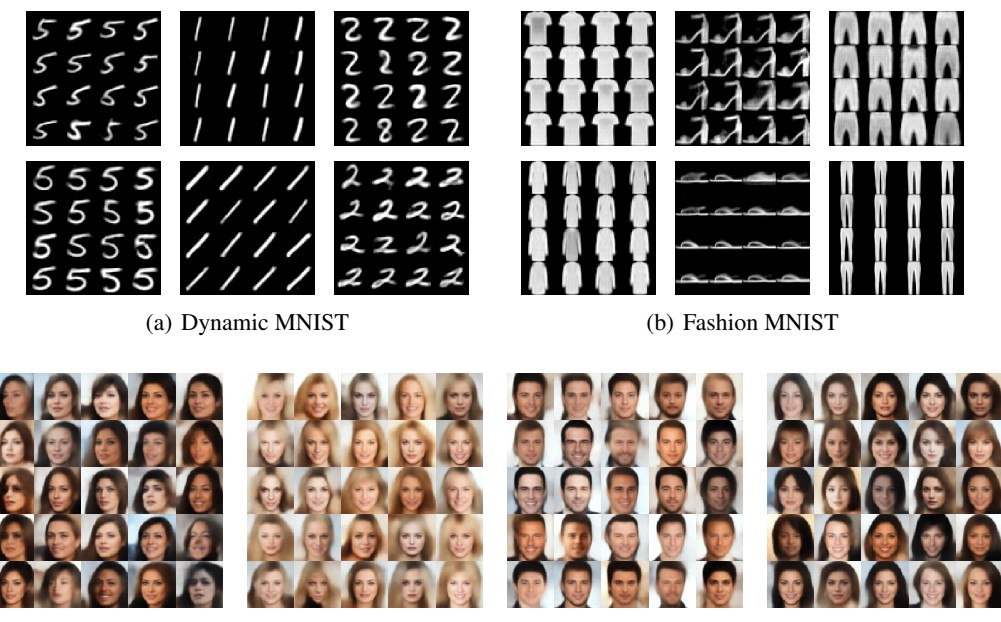

(a) Dynamic MNIST           (b) Fashion MNIST

(c) CelebA

Figure 1: Samples generated by ByPE-VAE based on the same pseudodata point in each plate. These show that ByPE-VAE can generate high-quality samples with various identifiable features of the data while inducing a cluster without losing diversity.

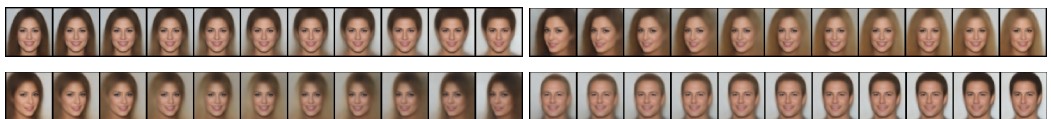

Figure 2: Interpolation between samples from the CelebA dataset.

prior for the latent representations of MNIST test data. The corresponding t-SNE visualization is shown in Fig.3. Test points with the same label are marked with the same color. For the latent representations of our model, the distance between classes is larger and the distance within classes is smaller. They are more meaningful than the representations of VAE. Then, we compare ByPE-VAE with the other three VAEs on two datasets for the k-nearest neighbor (kNN) classification task. Fig.4 shows the results for different values of $K$, where $K \in \{3, 5, 7, 9, 11, 13, 15\}$. ByPE-VAE consistently outperforms other models on MNIST and Fashion MNIST. Results on CIFAR10 are reported on Supp.F since the space constraints.

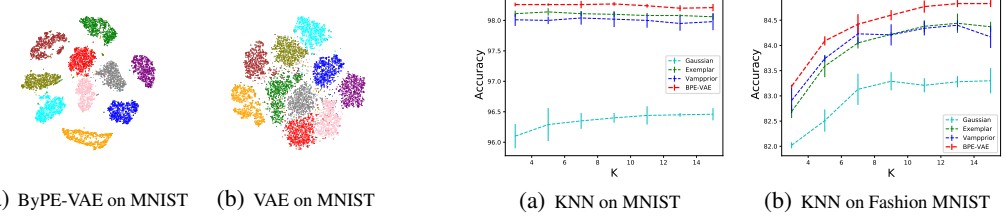

(a) ByPE-VAE on MNIST    (b) VAE on MNIST        (a) KNN on MNIST      (b) KNN on Fashion MNIST

Figure 3: t-SNE visualization of latent representations for test set, colored by labels.

Figure 4: kNN classification accuracy (%) with different values of $K$ on MNIST and Fashion MNIST.

## 5.3 Efficiency Analysis

We also compare ByPE-VAE with the Exemplar VAE from two aspects to further verify the efficiency of our model. First, we report the log-likelihood values of two models for the test set, as shown

| Dataset | ByPE VAE (500) | | Vamp VAE (500) | | Exemplar VAE (25000) | |
|---|---|---|---|---|---|---|
| | NLL | Time | NLL | Time | NLL | Time |
| Dynamic MNIST | 23.61 | 13.19 | 23.65 | 13.03 | 23.61 | 35.45 |
| Fashion MNIST | 20.85 | 12.05 | 20.87 | 12.09 | 20.81 | 37.23 |
| CIFAR10 | 71.91 | 17.30 | 71.97 | 14.89 | 72.00 | 66.85 |

Table 2: Resluts of average negative log-likelihood and training time (s/epoch) with various datasets.

in Fig. 5. In the case of the same size of exemplars and pseudocoreset, the performance of ByPE-VAE is significantly better than Exemplar VAE. Second, we record the average training time of two models for each epoch. Here, $M$ is set to 500, and the size of the exemplars is set to 25000, which is consistent with the value reported by Exemplar VAE. Note that in the part of the fixed pseudocoreset, the training time of ByPE-VAE is very small and basically the same as that of VAE with a Gaussian prior. As a result, the main time-consuming part of our model lies in the update of the pseudocoreset. However, the pseudocoreset needs to update every $k$ epochs only while $k$ is set to 10 in our experiments. All experiments are run on a single Nvidia 1080Ti GPU. The results can be seen in Table 2, where we find that our model obtains about $3\times$ speed-up while almost holding the performance.

## 5.4 Generative Data Augmentation

Finally, we evaluate the performance of ByPE-VAE for generating augmented data to further improve discriminative models. To be more comprehensive and fair, we adopt two ways to generate extra samples. The first way is to sample the latent representation from the posterior $q_\phi$ and then use it to generate a new sample for all models. The second way is to sample the latent representation from the prior $p_\phi$ and then use it to generate a new sample. This way is only applicable to the ByPE-VAE and the Exemplar VAE. Note that, it is a little different from the generation process of our method in this task. Due to the lack of labels in the pseudocoreset, we cannot directly let the prior $p_\phi$ be conditioned on the pseudocoreset. Specifically, we use the original training data to replace the pseudocoreset since the KL divergence between $p_\phi(\mathbf{z} \mid U, \mathbf{w})$ and $p_\phi(\mathbf{z} \mid X)$ has become small at the end of the training. Additionally, the generated samples are labeled by corresponding original training data. We train the discriminative model on a mixture of original training data and generated data. Each training iteration is as follows, which refers to section 5.4 of [13],

- Sample a minibatch $X = \{(\mathbf{x}_i, y_i)\}_{i=1}^{B}$ from training data.
- For each $\mathbf{x}_i \in X$, draw $\mathbf{z}_i \sim q_\phi(\mathbf{z} \mid \mathbf{x}_i)$ or $\mathbf{z}_i \sim r_\phi(\mathbf{z} \mid \mathbf{x}_i)$, which correspond to two ways respectively.
- For each $\mathbf{z}_i$, set $\tilde{\mathbf{x}}_i = p_\theta(\mathbf{x} \mid \mathbf{z}_i)$, which assigns the label $y_i$, and then obtain a synthetic minibatch $\tilde{X} = \{(\tilde{\mathbf{x}}_i, y_i)\}_{i=1}^{B}$.
- Optimize the weighted cross entropy: $\ell = -\sum_{i=1}^{B} [\lambda \log p_\theta(y_i \mid \mathbf{x}_i) + (1 - \lambda) \log p_\theta(y_i \mid \tilde{\mathbf{x}}_i)]$.

As reported in [13], the hyper-parameter $\lambda$ is set to $0.4$. The network architecture of the discriminative model is an MLP network with two hidden layers of 1024 units and ReLU activations are adopted. The results are summarized in Table 3. The test error of ByPE-VAE is lower than other models on MNIST for both sampling ways.

| Model | Test-error |
|---|---|
| Gaussian prior w/ Variational Posterior | $1.23 \pm 0.02$ |
| Vampprior w/ Variational Posterior | $1.20 \pm 0.02$ |
| Exemplar prior w/ Variational Posterior | $1.16 \pm 0.01$ |
| ByPE-VAE w/ Variational Posterior | $1.10 \pm 0.01$ |
| Exemplar prior w/ Prior | $1.10 \pm 0.01$ |
| ByPE-VAE w/ Prior | $\mathbf{0.88} \pm 0.02$ |

Table 3: Test error (%) on permutation invariant MNIST.

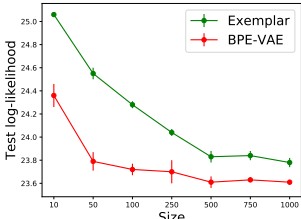

Figure 5: Average negative log-likelihood on test set with the different size of exemplars for Exemplar VAE and ByPE-VAE, respectively.

# 6  Conclusion

In this paper, we introduce ByPE-VAE, a new variant of VAE with a Bayesian Pseudocoreset based prior. The proposed prior is conditioned on a small-scale meaningful pseudocoreset rather than large-scale training data, which greatly reduces the computational complexity and prevents overfitting. Additionally, through the variational inference formulation, we obtain the optimal pseudocoreset to approximate the entire dataset. For optimization, we employ a two-step alternative search strategy to optimize the parameters in the VAE framework and the pseudodata points along with weights in the pseudocoreset. Finally, we demonstrate the promising performance of the ByPE-VAE in a number of tasks and datasets.

## Acknowledgment

This paper was partially supported by the National Key Research and Development Program of China (No. 2018AAA0100204), and a key program of fundamental research from Shenzhen Science and Technology Innovation Commission (No. JCYJ20200109113403826). We thank Dr. Liangjian Wen for an insightful discussion on the design of our frameworks.

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
