# A  Derivation of Eq. (9)

$$\log p(\mathbf{x}; U, \mathbf{w}, \theta, \phi) = \log \sum_{m=1}^{M} \frac{w_m}{N} T_{\phi,\theta}(\mathbf{x} \mid \mathbf{u}_m)$$

$$= \log \sum_{m=1}^{M} \frac{w_m}{N} \int_z r_\phi(\mathbf{z} \mid \mathbf{u}_m) \, p_\theta(\mathbf{x} \mid \mathbf{z}) d\mathbf{z}$$

$$= \log \int_z p_\theta(\mathbf{x} \mid \mathbf{z}) \sum_{m=1}^{M} \frac{w_m}{N} r_\phi(\mathbf{z} \mid \mathbf{u}_m) \, d\mathbf{z}$$

$$= \log \int_z \frac{q_\phi(\mathbf{z} \mid \mathbf{x}) p_\theta(\mathbf{x} \mid \mathbf{z}) \sum_{m=1}^{M} w_m r_\phi(\mathbf{z} \mid \mathbf{u}_m) / N}{q_\phi(\mathbf{z} \mid \mathbf{x})} d\mathbf{z}$$

$$\geq \mathop{\mathbb{E}}_{q_\phi(\mathbf{z}|\mathbf{x})} \log p_\theta(\mathbf{x} \mid \mathbf{z}) - \mathop{\mathbb{E}}_{q_\phi(\mathbf{z}|\mathbf{x})} \log \frac{q_\phi(\mathbf{z} \mid \mathbf{x})}{\sum_{m=1}^{M} w_m r_\phi(\mathbf{z} \mid \mathbf{u}_m) / N}$$

$$= O(\theta, \phi, U, \mathbf{w}; \mathbf{x}). \tag{20}$$

# B  Derivations of Eqs. (17) - (19)

## B.1  Derivation of KL divergence in Eq. (17)

$$D_{\mathrm{KL}}\left(p_\phi(\mathbf{z} \mid U, \mathbf{w}) \| p_\phi(\mathbf{z} \mid X)\right)$$

$$= \int_z p_\phi(\mathbf{z} \mid U, \mathbf{w}) \left[\log p_\phi(\mathbf{z} \mid U, \mathbf{w}) - \log p_\phi(\mathbf{z} \mid X)\right] d\mathbf{z}$$

$$= \mathbb{E}_{U,\mathbf{w}}[\log p_\phi(\mathbf{z} \mid U, \mathbf{w}] - \mathbb{E}_{U,\mathbf{w}}[\log p_\phi(\mathbf{z} \mid X)]$$

$$= \mathbb{E}_{U,\mathbf{w}}\left[-\log Z(U, \mathbf{w}) + \sum_{m=1}^{M} w_m \log p_\theta(\mathbf{u}_m \mid \mathbf{z})\right] - \mathbb{E}_{U,\mathbf{w}}\left[-\log Z(\mathbf{1}_N) + \sum_{n=1}^{N} 1 \times \log p_\theta(\mathbf{x}_n \mid \mathbf{z})\right]$$

$$= \log Z(\mathbf{1}_N) - \log Z(U, \mathbf{w}) - \mathbf{1}_N^T \mathbb{E}_{U,\mathbf{w}}[\log p_\theta(X \mid \mathbf{z})] + \mathbf{w}^T \mathbb{E}_{U,\mathbf{w}}[\log p_\theta(U \mid \mathbf{z})]. \tag{21}$$

## B.2  Derivation of Eq. (18)

The gradient of Eq. (21) with respect to a single pseudopoint $\mathbf{u}_m \in \mathbb{R}^d$ can be expressed by

$$\nabla_{\mathbf{u}_m} D_{\mathrm{KL}} = -\nabla_{\mathbf{u}_m} \log Z(U, \mathbf{w}) - \nabla_{\mathbf{u}_m} \mathbb{E}_{U,\mathbf{w}}\left[(\log p_\theta(X \mid \mathbf{z}))^T \mathbf{1}_N\right] + \nabla_{\mathbf{u}_m} \mathbb{E}_{U,\mathbf{w}}\left[(\log p_\theta(U \mid \mathbf{z}))^T \mathbf{w}\right]. \tag{22}$$

First, we compute the gradient of the log normalization constant $\nabla_{\mathbf{u}_m} \log Z(U, \mathbf{w})$ by

$$\nabla_{\mathbf{u}_m} \log Z(U, \mathbf{w}) = \frac{1}{Z(U, \mathbf{w})} \nabla_{\mathbf{u}_m} \int \exp\left(\mathbf{w}^T \log p_\theta(U \mid \mathbf{z})\right) p_0(\mathbf{z}) d\mathbf{z}$$

$$= \int \frac{1}{Z(U, \mathbf{w})} p_0(\mathbf{z}) \nabla_{\mathbf{u}_m} \left(\exp\left(\mathbf{w}^T \log p_\theta(U \mid \mathbf{z})\right)\right) d\mathbf{z}$$

$$= \int \frac{1}{Z(U, \mathbf{w})} p_0(\mathbf{z}) \exp\left(\mathbf{w}^T \log p_\theta(U \mid \mathbf{z})\right) \nabla_{\mathbf{u}_m} \left(\mathbf{w}^T \log p_\theta(U \mid \mathbf{z})\right) d\mathbf{z}$$

$$= w_m \mathbb{E}_{U,\mathbf{w}}\left[\nabla_{\mathbf{u}_m} \log p_\theta(\mathbf{u}_m \mid \mathbf{z})\right]. \tag{23}$$

Then, for any function $a(U, \mathbf{z}) : \mathbb{R}^{d \times M} \times \mathbf{Z} \to \mathbb{R}$, we have

$$\nabla_{\mathbf{u}_m} \mathbb{E}_{U,\mathbf{w}}\left[a(U, \mathbf{z})\right] = \int \nabla_{\mathbf{u}_m} \left(\exp\left(\mathbf{w}^T \log p_\theta(U \mid \mathbf{z}) - \log Z(U, \mathbf{w})\right) a(U, \mathbf{z})\right) p_0(\mathbf{z}) d\mathbf{z}. \tag{24}$$

Using the product rule,

$$\nabla_{\mathbf{u}_m} \mathbb{E}_{U,\mathbf{w}}\left[a(U, \mathbf{z})\right]$$
$$= \mathbb{E}_{U,\mathbf{w}}\left[\nabla_{\mathbf{u}_m} a(U, \mathbf{z})\right] + \mathbb{E}_{U,\mathbf{w}}\left[a(U, \mathbf{z})(w_m \nabla_{\mathbf{u}_m} \log p_\theta(\mathbf{u}_m \mid \mathbf{z})) - \nabla_{\mathbf{u}_m} \log Z(U, \mathbf{w})\right]. \tag{25}$$

Combining Eq. (23) and Eq. (25), we have

$$\nabla_{\mathbf{u}_m} \mathbb{E}_{U,\mathbf{w}} \left[ a(U, \mathbf{z}) \right]$$
$$= \mathbb{E}_{U,\mathbf{w}} \left[ \nabla_{\mathbf{u}_m} a(U, \mathbf{z}) \right] + w_m \mathbb{E}_{U,\mathbf{w}} \left[ a(U, \mathbf{z}) \left( \nabla_{\mathbf{u}_m} \log p_\theta(\mathbf{u}_m \mid \mathbf{z}) - \mathbb{E}_{U,\mathbf{w}} \left[ \nabla_{\mathbf{u}_m} \log p_\theta(\mathbf{u}_m \mid \mathbf{z}) \right] \right) \right]. \tag{26}$$

Subtracting $0 = \mathbb{E}_{U,\mathbf{w}} \left[ a(U, \mathbf{z}) \right] \mathbb{E}_{U,\mathbf{w}} \left[ \left( \nabla_{\mathbf{u}_m} \log p_\theta(\mathbf{u}_m \mid \mathbf{z}) - \mathbb{E}_{U,\mathbf{w}} \left[ \nabla_{\mathbf{u}_m} \log p_\theta(\mathbf{u}_m \mid \mathbf{z}) \right] \right) \right]$ yields

$$\nabla_{\mathbf{u}_m} \mathbb{E}_{U,\mathbf{w}} \left[ a(U, \mathbf{z}) \right] = \mathbb{E}_{U,\mathbf{w}} \left[ \nabla_{\mathbf{u}_m} a(U, \mathbf{z}) \right] + w_m \, \mathrm{Cov} \left[ a(U, \mathbf{z}), \nabla_{\mathbf{u}_m} \log p_\theta(\mathbf{u}_m \mid \mathbf{z}) \right]. \tag{27}$$

Finally, the gradient with respect to $\mathbf{u}_i$ in Eq. (18) obtains by substituting $(\log p_\theta(X \mid \mathbf{z}))^T \mathbf{1}_N$ and $(\log p_\theta(U \mid \mathbf{z}))^T \mathbf{w}$ for $a(U, \mathbf{z})$.

### B.3 Derivations of Eq. (19)

Similar to derivation above, we give the gradient with respect to weight vector $\mathbf{w} \in \mathbb{R}_+^M$, which is given by

$$\nabla_{\mathbf{w}} \mathrm{D}_{\mathrm{KL}} = -\nabla_{\mathbf{w}} \log Z(U, \mathbf{w}) - \nabla_{\mathbf{w}} \mathbb{E}_{U,\mathbf{w}} \left[ (\log p_\theta(X \mid \mathbf{z}))^T \mathbf{1}_N \right] + \nabla_{\mathbf{w}} \mathbb{E}_{U,\mathbf{w}} \left[ (\log p_\theta(U \mid \mathbf{z}))^T \mathbf{w} \right]. \tag{28}$$

First, we compute the gradient of the log normalization constant via

$$\nabla_{\mathbf{w}} \log Z(U, \mathbf{w}) = \int \frac{1}{Z(U, \mathbf{w})} \nabla_{\mathbf{w}} \left( \exp \left( \mathbf{w}^T \log p_\theta(U \mid \mathbf{z}) \right) \right) p_0(\mathbf{z}) d\mathbf{z}$$
$$= \int \frac{1}{Z(U, \mathbf{w})} p_0(\mathbf{z}) \exp \left( \mathbf{w}^T \log p_\theta(U \mid \mathbf{z}) \right) \nabla_{\mathbf{w}} \left( \mathbf{w}^T \log p_\theta(U \mid \mathbf{z}) \right) d\mathbf{z}$$
$$= \mathbb{E}_{U,\mathbf{w}} \left[ \log p_\theta(U \mid \mathbf{z}) \right]. \tag{29}$$

Then, for any function $a : \mathbf{Z} \to \mathbb{R}$, we have

$$\nabla_{\mathbf{w}} \mathbb{E}_{U,\mathbf{w}} \left[ a(\mathbf{z}) \right] = \nabla_{\mathbf{w}} \int \left( \exp \left( \mathbf{w}^T \log p_\theta(U \mid \mathbf{z}) - \log Z(U, \mathbf{w}) \right) \right) a(\mathbf{z}) p_0(\mathbf{z}) d\mathbf{z}$$
$$= \int \nabla_{\mathbf{w}} \left( \exp \left( \mathbf{w}^T \log p_\theta(U \mid \mathbf{z}) - \log Z(U, \mathbf{w}) \right) \right) p_0(\mathbf{z}) a(\mathbf{z}) d\mathbf{z}$$
$$= \mathbb{E}_{U,\mathbf{w}} \left[ (\log p_\theta(U \mid \mathbf{z}) - \nabla_{\mathbf{w}} \log Z(U, \mathbf{w})) \, a(\mathbf{z}) \right]. \tag{30}$$

Combining Eq. (29) and Eq. (30), we have

$$\nabla_{\mathbf{w}} \mathbb{E}_{U,\mathbf{w}} \left[ a(\mathbf{z}) \right] = \mathbb{E}_{U,\mathbf{w}} \left[ (\log p_\theta(U \mid \mathbf{z}) - \mathbb{E}_{U,\mathbf{w}} \left[ \log p_\theta(U \mid \mathbf{z}) \right]) \, a(\mathbf{z}) \right]. \tag{31}$$

Subtracting $0 = \mathbb{E}_{U,\mathbf{w}}[a(\mathbf{z})] \mathbb{E}_{U,\mathbf{w}} \left[ \log p_\theta(U \mid \mathbf{z}) - \mathbb{E}_{U,\mathbf{w}} \left[ \log p_\theta(U \mid \mathbf{z}) \right] \right]$ yields

$$\nabla_{\mathbf{w}} \mathbb{E}_{U,\mathbf{w}}[a(\mathbf{z})] = \mathrm{Cov} \left[ \log p_\theta(U \mid \mathbf{z}), a(\mathbf{z}) \right]. \tag{32}$$

Using the product rule, the gradient with respect to $\mathbf{w}$ in Eq. (19) follows by substituting $\mathbf{1}_N^T \log p(X \mid \mathbf{z})$ and $\mathbf{w}^T \log p_\theta(U \mid \mathbf{z})$ for $a(\mathbf{z})$.

## C  Derivation of Algorithm 2

First, We initialize the pseudocoreset through subsampling $M$ datapoints from the whole dataset and reweighting them to match the overall weight of the full dataset,

$$\mathbf{u}_m \leftarrow \mathbf{x}_{b_m}, \quad w_m \leftarrow N/M, \quad m = 1, \dots, M$$
$$\mathcal{B} \sim \mathrm{UnifSubset} \left( [N], M \right), \quad \mathcal{B} := \{ b_1, \dots, b_M \}.$$

After initializing, we simultaneously optimize Eq. (17) over both pseudodata points and weights. The learning rate of each stochastic gradient descent step is $\gamma_t \propto t^{-1}$, where $t \in \{1, \cdots, T\}$ denotes the iteration for optimization. Then,

$$w_m \leftarrow \max \left( 0, w_m - \gamma_t \left( \hat{\nabla}_w \right)_m \right), \quad \mathbf{u}_m \leftarrow \mathbf{u}_m - \gamma_t \hat{\nabla}_{\mathbf{u}_m}, \quad 1 \le m \le M \tag{33}$$

where $\hat{\nabla}_{\mathbf{w}} \in \mathbb{R}^M$ and $\hat{\nabla}_{\mathbf{u}_m} \in \mathbb{R}^d$ are the stochastic gradients of $\mathbf{w}$ and $\mathbf{u}_m$ respectively. Based on $S \in \mathbb{N}$ samples $(\mathbf{z})_{s=1}^S \sim p_\phi(\mathbf{z}|U, \mathbf{w})$ from the coreset approximation and a minibatch of $B \in \mathbb{N}$ datapoints from the full dataset, we obtain these stochastic gradients, as follows,

$$\hat{\nabla}_{\mathbf{w}} = -\frac{1}{S}\sum_{s=1}^S \tilde{\mathbf{g}}_s\left(\frac{N}{B}\mathbf{g}_s^T 1 - \tilde{\mathbf{g}}_s^T \mathbf{w}\right), \quad \hat{\nabla}_{\mathbf{u}_m} = -w_m \frac{1}{S}\sum_{s=1}^S \tilde{\mathbf{h}}_{m,s}\left(\frac{N}{B}\mathbf{g}_s^T 1 - \tilde{\mathbf{g}}_s^T \mathbf{w}\right), \quad (34)$$

where,

$$\tilde{\mathbf{h}}_{m,s} = \nabla_U \log p_\theta(\mathbf{u}_m|\mathbf{z}_s) - 1/S \sum_{s'=1}^S \nabla_U \log p_\theta(\mathbf{u}_m|\mathbf{z}_{s'})), \quad (35)$$

$$\mathbf{g}_s = \left(\log p_\theta(\mathbf{x}_b|\mathbf{z}_s) - 1/S \sum_{s'=1}^S \log p_\theta(\mathbf{x}_b|\mathbf{z}_{s'})\right)_{b\in\mathcal{B}}, \quad (36)$$

$$\tilde{\mathbf{g}}_s = \left(\log p_\theta(\mathbf{u}_m|\mathbf{z}_s) - 1/S \sum_{s'=1}^S \log p_\theta(\mathbf{u}_m|\mathbf{z}_{s'})\right)_{m=1}^M. \quad (37)$$

This process is shown in Algorithm 2.

## D  More t-SNE visualization results

We already report the t-SNE visualization of ByPE-VAE and standard VAE in Figure. 3. Here we give more t-SNE visualization results.

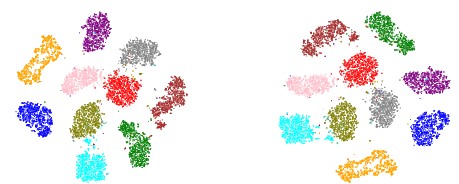

(a) VampPrior on MNIST   (b) Exemplar on MNIST

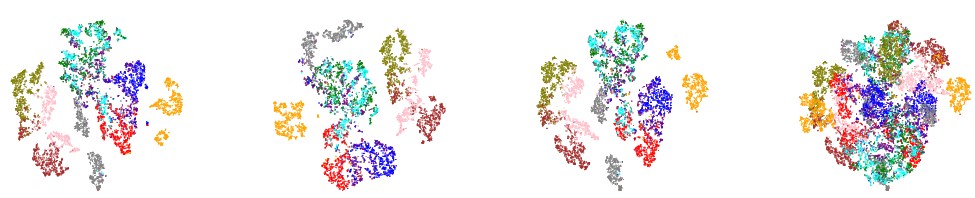

(c) ByPE-VAE on Fashion (d) VampPrior on Fashion (e) Exemplar on Fashion (f)  VAE    on    Fashion
MNIST                    MNIST                    MNIST                    MNIST

Figure 6: t-SNE visualization of learned latent representations, colored by labels.

## E  ByPE-VAE samples

First, we randomly sample from ByPE-VAEs trained on different datasets, namely, MNIST, Fashion MNIST, and Celeba, as shown in Fig.7. Second, we give more generated samples in Fig.8, among which the samples in each plate are based on the same pseudodata point.

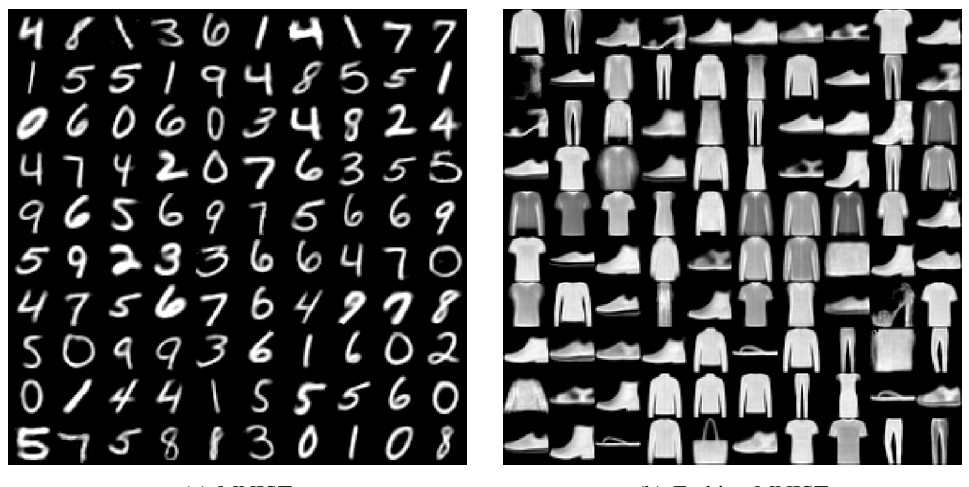

(a) MNIST

(b) Fashion MNIST

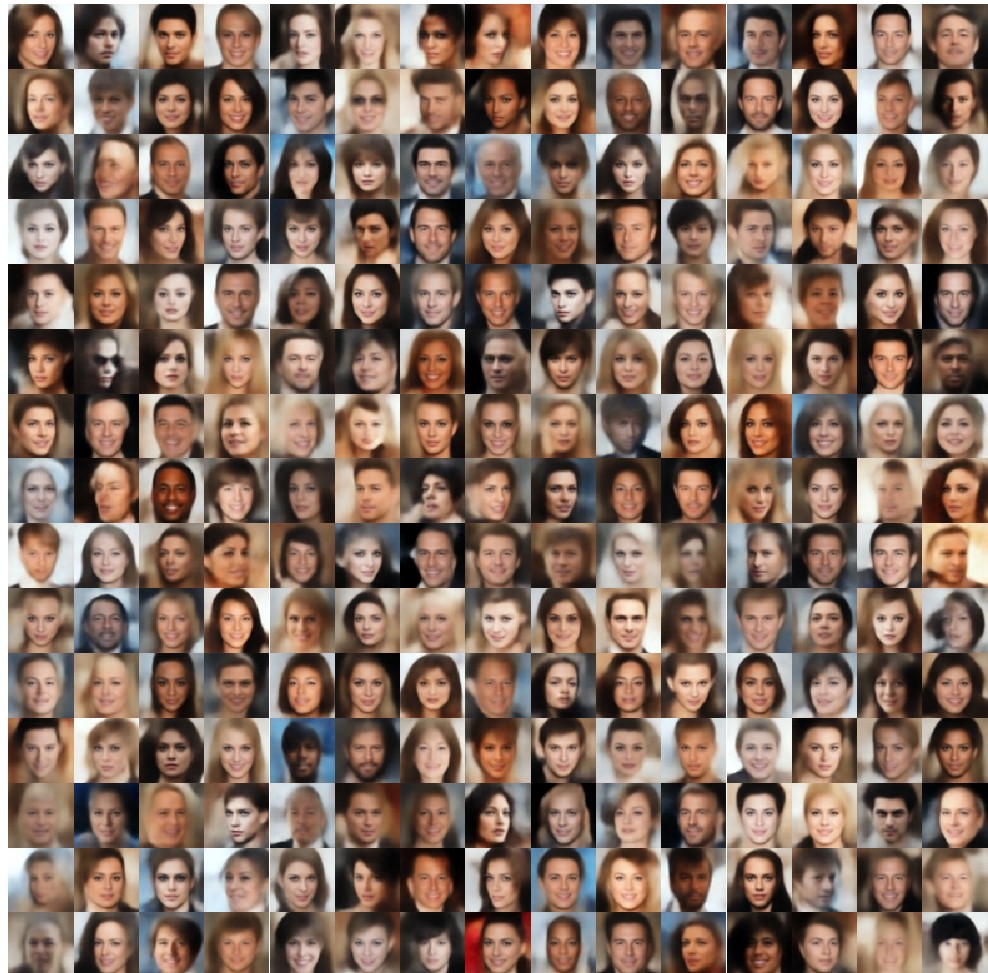

(c) CelebA

Figure 7: Random samples drawn from ByPE-VAEs trained on different datasets.

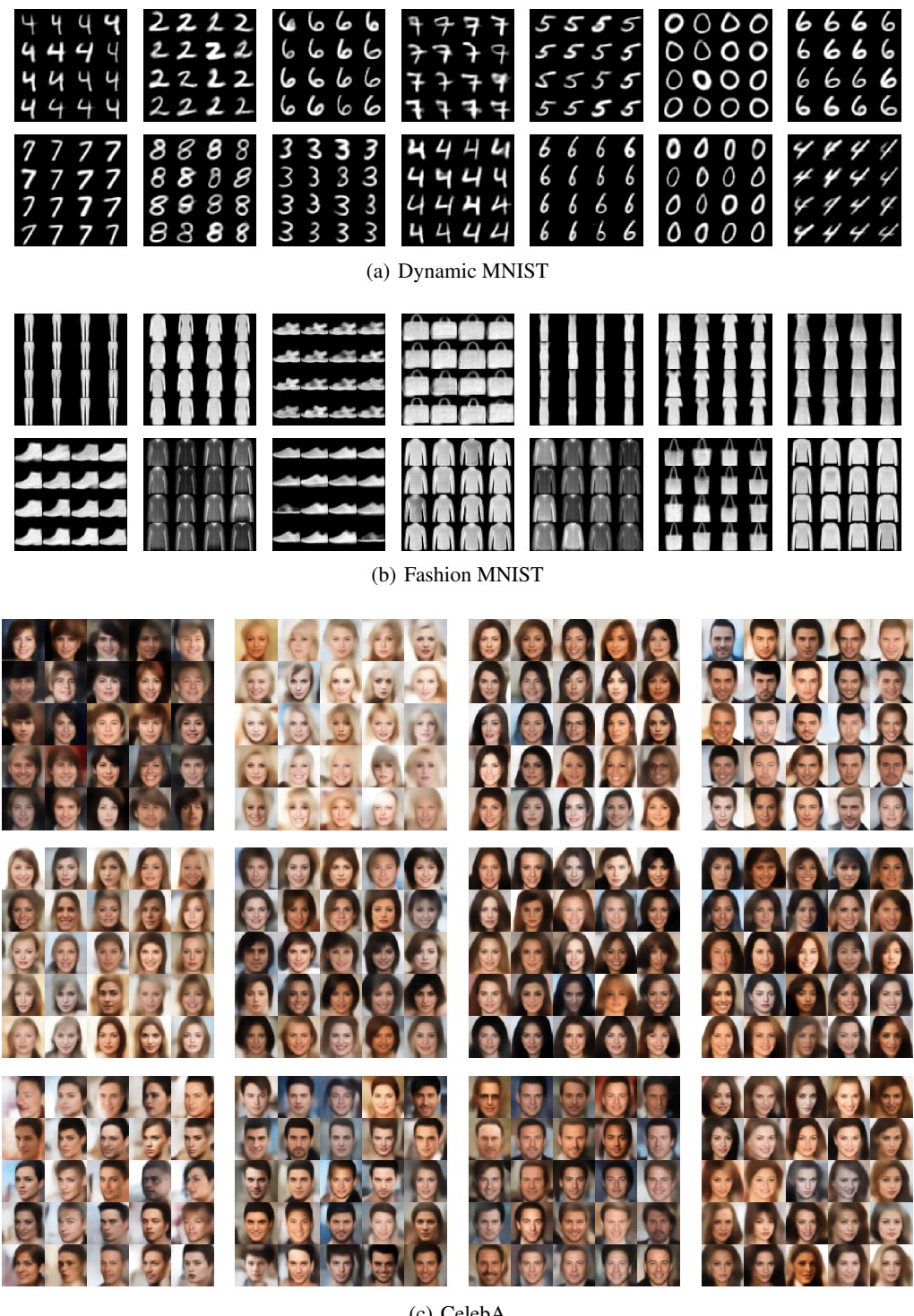

(a) Dynamic MNIST

(b) Fashion MNIST

(c) CelebA

Figure 8: Samples generated by ByPE-VAE based on the same pseudodata point in each plate.

## F  KNN on CIFAR10

In section 5.2, We only report the KNN results of MNIST and Fashion MNIST in the Fig. 4. Here we give the KNN results on Cifar10. As shown in Fig. 9, the results of ByPE-VAE are significantly better than other models with different values of $K \in \{3, 5, 7, 9, 11, 13, 15\}$.

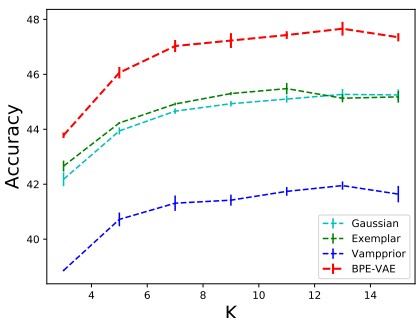

Figure 9: KNN on CIFAR10

## G  Interpolation between samples in CelebA

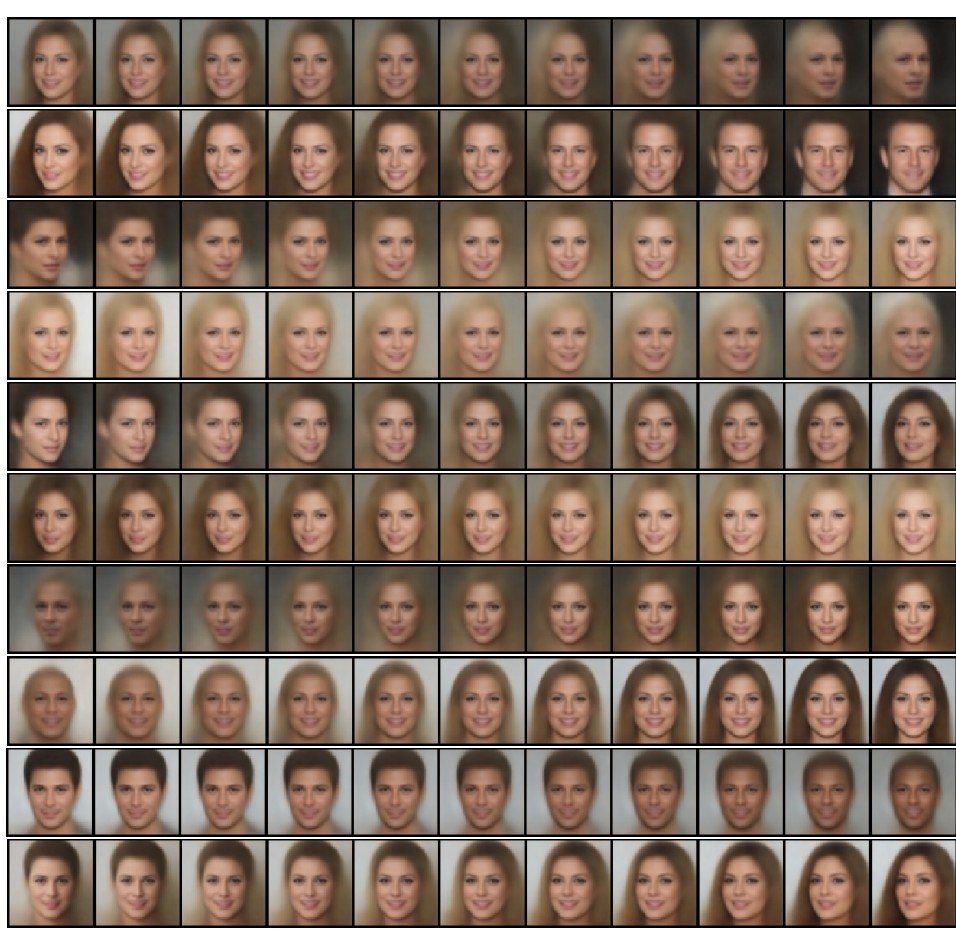

Figure 10: Interpolation between samples from the CelebA dataset.

## H   Density estimation on CelebA

We report the density estimation results on Dynamic MNIST, Fashion MNIST, and CIFAR10 based on different network architectures in Table. 1. Here we also report the test negative log-likelihood(NLL) for CelebA, as shown in the Table. 4 below. The experimental results show that ByPE-VAE outperforms other models.

| Method | Gaussian prior | VampPrior | Exemplar | ByPE |
|---|---|---|---|---|
| Test-loglikelihood | $183.16 \pm 0.40$ | $183.61 \pm 0.69$ | $185.03 \pm 1.46$ | $\mathbf{182.11} \pm 1.10$ |

Table 4: Density estimation on CelebA based on the Fully Convolutional Neural Network

## I   Sensitivity analysis on k

In our optimization algorithm, the pseudocoreset $\{U, \mathbf{w}\}$ is updated by every $k$ epochs rather than be updated every epoch. So, we also test the sensitivity about $k$. The results are summarized in Table. 5. Considering performance and time consumption, we set $k$ to 10 in the experiments.

| $k$ | $k = 1$ | $k = 10$ | $k = 50$ | $k = 100$ |
|---|---|---|---|---|
| ByPE-VAE on MNIST | 23.60 | 23.61 | 23.62 | 23.70 |

Table 5: Test negative log-likelihood on different update interval $k$

## J   More results on Generative Data Augmentation

In section 5.4, we report the test error on permutation invariant MNIST in Table. 3. Here we give the test error on permutation invariant Fashion MNIST and CIFAR10. The results are summarized in Table. 6. The test error of ByPE-VAE is lower than other models on most case for both sampling way.

| Model | Fashion MNIST | CIFAR10 |
|---|---|---|
| Gaussian prior w/ Variational Posterior | $9.98 \pm 0.08$ | $50.07 \pm 0.23$ |
| Vampprior w/ Variational Posterior | $10.03 \pm 0.05$ | $50.74 \pm 0.16$ |
| Exemplar prior w/ Variational Posterior | $\mathbf{9.46} \pm 0.02$ | $49.59 \pm 0.21$ |
| ByPE-VAE w/ Variational Posterior | $9.75 \pm 0.02$ | $\mathbf{49.00} \pm 0.04$ |
| Exemplar prior w/ Prior | $9.58 \pm 0.01$ | $47.20 \pm 0.13$ |
| ByPE-VAE w/ Prior | $\mathbf{9.56} \pm 0.02$ | $\mathbf{46.60} \pm 0.13$ |

Table 6: Test error (%) on permutation invariant Fashion MNIST and CIFAR10.

Then, we report the performance of our method based on different values of $\lambda$ in Table. 7. We use 0.4 as reported in [13].

## K   KL Loss

To measure these two different pseudo-inputs, we could compare the value of the KL divergence between the prior distribution and the variational posterior distribution. The results (shown in Table. 8) show our method mostly outperforms VampPrior on three datasets, indicating that the pseudo-inputs learned by our method are better.

| $\lambda$ | 0 | 0.1 | 0.2 | 0.3 | 0.4 | 0.5 |
|---|---|---|---|---|---|---|
| Test error | $1.42 \pm 0.08$ | $1.16 \pm 0.10$ | $0.93 \pm 0.01$ | $0.96 \pm 0.01$ | $0.88 \pm 0.02$ | $\mathbf{0.83}\pm 0.05$ |
| $\lambda$ | 0.6 | 0.7 | 0.8 | 0.9 | 1.0 | - |
| Test error | $0.90 \pm 0.02$ | $0.94 \pm 0.02$ | $0.98 \pm 0.01$ | $1.06 \pm 0.01$ | $1.31 \pm 0.00$ | - |

Table 7: MNIST test error versus $\lambda$, which controls the relative balance of real and augmented data

| KL | Dynamic MNIST | Fashion MNIST | CIFAR10 |
|---|---|---|---|
| VAE w/ VampPrior | $12.08 \pm 0.05$ | $8.05 \pm 0.05$ | $21.80 \pm 0.07$ |
| ByPE VAE | $\mathbf{11.93}\pm 0.12$ | $\mathbf{8.03}\pm0.01$ | $\mathbf{21.55}\pm 0.02$ |
| HVAE w/ VampPrior | $12.20 \pm 0.06$ | $8.15 \pm 0.03$ | $22.34 \pm 0.14$ |
| ByPE HVAE | $\mathbf{12.18}\pm 0.06$ | $\mathbf{8.10}\pm 0.04$ | $\mathbf{22.16}\pm 0.06$ |
| ConvHVAE w/ VampPrior | $12.66 \pm 0.06$ | $8.54 \pm 0.05$ | $24.42 \pm 0.24$ |
| ByPE ConvHVAE | $\mathbf{12.50}\pm 0.06$ | $\mathbf{8.49}\pm 0.05$ | $\mathbf{24.15}\pm 0.28$ |

Table 8: The comparision of KL loss on different datasets

## L   Dynamics of optimization process

To better examine the dynamics of the two-stage optimization approach, we drawn the negative log-likelihood curve on validation set. As shown in Figure. 11, the loss function is steadily decreasing, except for an increase in the first update of pseudocoreset, and is convergent at the end.

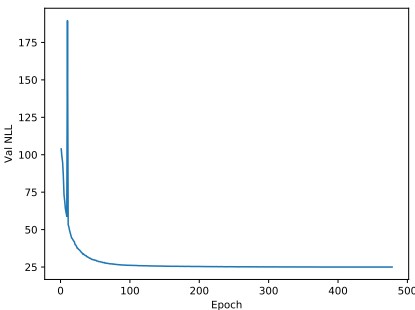

Figure 11: Loss curve of ByPE-VAE on MNIST validation set.

## M   Hyper-parameters in Experiments

For each dataset, we use a 40-dimensional latent space. We use Gradient Normalized Adam with learning rate of $5e-4$ and minibatch size of 100 for all of the datasets. For the sake of uniformity, all data sets are continuous, that is, the pixel value is compressed to between 0 and 1. We use early-stopping with a look ahead of 50 epochs to stop training. That is, if for 50 consecutive epochs the validation ELBO does not improve, we stop the training process. The gating mechanism is used for all activation functions. The size of pseudocoresets is 500 for all experiments except 240 for CelebA. The stepsize used in pseudocoresets updating is best in $\{0.1, 0.5\}$. The update interval $k$ of pseudocoresets is 10. All results are averaged over 3 random training runs.