# OpenReview forum: "ByPE-VAE: Bayesian Pseudocoresets Exemplar VAE"
_NeurIPS.cc/2021/Conference — NeurIPS 2021 Poster_

### Official Review · Reviewer_LzjC · 2021-07-12

**Rating:** 6
**Confidence:** 4

**Summary:**

This paper proposes the use of pseudo-coresets to accelerate inference in Examplar VAE.
The Examplar VAE ELBO (Eq.4/5 in [ExamplarVAE]) is updated to use pseudo-coreset (Eq.8/9 in this paper). Other quantities of interest are updated accordingly and a two-stage approach is proposed to update two types of parameters (VAE and pseudo-coresets).

[ExamplarVAE] Norouzi et al (2020) "Exemplar VAE: Linking Generative Models, Nearest Neighbor Retrieval, and Data Augmentation", NeurIPS 2020.

**Ethical Concerns:**

-

**Limitations And Societal Impact:**

-

**Main Review:**

The authors do not report the sensitivity for the parameter 'k' (pseudocoreset are updated every 'k' epochs).
It would be interesting to show a few different values to understand how important these updates are, for example how data-dependent.
In KNN the number of neighboors is also 'K' which can be confusing.

The dynamics of the two-stage optimization approach are not examined. This would be important as it is a key novel component of the proposed method. Is convergence stable across experimental settings? does it requires any tuning?

Fig.5: are the differences in log-likelihood statistically significant?

Overall, the goal is clear and the methodology is rigorously detailed, including all equations of interests.
My main concern is novelty: the method is a relatively straightforward extension of Examplar VAE using Pseudo-coresets - both of these techniques are pre-existent.
Additionally, it would have been good to demonstrate this approach on a larger/more challenging dataset.

**Time Spent Reviewing:**

2

---

> ### Author Response · Authors · 2021-08-09
> **Response to reviewer LzjC**
>
> **Q1.** *The authors do not report the sensitivity for the parameter 'k' (pseudocoreset are updated every 'k' epochs). It would be interesting to show a few different values to understand how important these updates are, for example how data-dependent. In KNN the number of neighboors is also 'K' which can be confusing.*
>
> **R.** Thanks for your valuable comments. We evaluate the sensitivity for the parameter `k' with different values. The experimental results are shown below(Table.9). And we use the different notation to denote the the number of neighboors in KNN.
>
> | k                 | k=1   | k=10  | k=50  | k=100 |
> | ----------------- | ----- | ----- | ----- | ----- |
> | ByPE VAE on MNIST | 23.60 | 23.61 | 23.62 | 23.70 |
>
> Table 9. Test negative log-likelihood on different update interval k
>
>
>
> **Q2.** *The dynamics of the two-stage optimization approach are not examined. This would be important as it is a key novel component of the proposed method. Is convergence stable across experimental settings? does it requires any tuning?*
>
> **R.** Thanks for your thoughtful feedback. We plot the curve of loss function during training, which can be seen in the revised manuscript. Due to the template, we cannot display the figure here.  It can be seen from the curve that the loss function is steadily decreasing, except for an increase in the first update of coreset, and is convergent at the end.
>
>
>
> **Q3.** *Fig.5: are the differences in log-likelihood statistically significant?*
>
> **R.**  Thanks for pointing this out. We also record the statistical significance of  negative loglikelihood on test set with the different size of exemplars in Fig.5. Due to the template, we can't display figures here, so we use the following table instead. In the revised version, we will modify Fig. 5 to better display.
>
> | k        | 10             | 50             | 100           | 250            | 500            | 750            | 1000           |
> | -------- | -------------- | -------------- | ------------- | -------------- | -------------- | -------------- | -------------- |
> | exemplar | 25.06+/- 0.02  | 24.54 +/- 0.06 | 24.28+/- 0.03 | 24.05+/- 0.02  | 23.83 +/- 0.04 | 23.83 +/- 0.04 | 23.78 +/- 0.04 |
> | ByPE     | 24.36 +/- 0.20 | 23.79 +/- 0.08 | 23.72+/- 0.08 | 23.71 +/- 0.06 | 23.61+/- 0.03  | 23.63 +/- 0.02 | 23.61 +/- 0.01 |
>
> Table 10. Average negative loglikelihood on test set with the different size of exemplars for Exemplar VAE and ByPE-VAE, respectively.
>
>
>
> **Q4.** *Overall, the goal is clear and the methodology is rigorously detailed, including all equations of interests. My main concern is novelty: the method is a relatively straightforward extension of Examplar VAE using Pseudo-coresets - both of these techniques are pre-existent. Additionally, it would have been good to demonstrate this approach on a larger/more challenging dataset.*
>
> **R.** Our method is not a simple extension of the Exemplar-VAE. First, we find the limit of Examplar-VAE which usually requires vast amounts of data to participate in training leading to huge computational complexity. Then, facing this problem, we introduce Bayesian pseudocoresets. In terms of theory, pseudocoresets can integrate well with the training process of VAE. In terms of experimental results, compared to the Exemplar VAE, ByPE-VAE is up to 3 times speed-up without losing performance on Dynamic MNIST, Fashion MNIST, and CIFAR10.  And we have evaluated our method on CelebA, which is considered as a large dataset with 202,599 face pictures.

---

### Official Review · Reviewer_rLyr · 2021-07-16

**Rating:** 6
**Confidence:** 4

**Summary:**

The paper proposes a Bayesian pseudocoreset based exemplar VAE algorithm which utilizes pseudocoresets to improve the computational complexity and reduce over-fitting of Exemplar-VAE (a related work), by conditioning the prior on a subset of data. The paper proposes variational inference to find the optimal pseudocoreset, by minimizing the KL divergence between the posteriors defined on the pseudocoreset and the training data. Detailed experimental evaluation is provided against baselines and competitors and shows improved performance of the proposed approach in running time and quality of the learned latent space.

The main contributions of the paper are:
- Bayesian pseudocoresets proposed instead of using all the training data as exemplars in  Exemplar-VAE
- Simultaneous optimization of the optimal pseudocoreset and VAE training, with the optimal pseudocoreset selected through variational inference
- Detailed experimental evaluation against competitors showing improved running time and latent space quality.

**Limitations And Societal Impact:**

Limitations are addressed, no negative societal impact observed.

**Main Review:**

The paper proposes Bayesian pseudocoresets to improve the computational complexity of Exemplar-VAE and to reduce overfitting.  The paper is well-written, and extensive experimental evaluation is provided. Variational inference to find optimal pseudocoresets, and the joint optimization along with VAE are strong technical contributions. However, the paper very closely follows the structure, formatting, and experimental evaluation of Exemplar-VAE (a related work). New experimental evaluation or insights on the pseudocoresets are not provided but a repetition of the same experiments conducted by Exemplar-VAE on the proposed approach. The novelty of the paper is thus somewhat limited and incremental. The pseudocode for the algorithm optimization is not explained in the text.

Detailed comments:

- The optimization process in Algorithm 2 is not explained in the paper. What is the significance of subsets S and M in Algorithm 2?
- Equations (9)-(14) are similar or same as the ones in Exemplar-VAE. The equations defined in Exemplar-VAE should be cited.
- Statistical significance of results on KNN is not provided. The results on density estimation show marginal improvement for the proposed approach.
- The t-SNE plots should also compare the latent space of Exemplar-VAE. Currently only the proposed approach and a baseline VAE are compared.
- Experiments to show how well the learned pseudocoresets approximate the training data, would be a useful addition.
- A comparison of KNN/log-likelihood for different coreset sizes would be useful. How does the proposed approach perform if the coreset size is set equal to the exemplar size in Exemplar-VAE?





**Time Spent Reviewing:**

2.5 hours

---

> ### Author Response · Authors · 2021-08-09
> **Response to reviewer rLyr**
>
> **Q1.** *The paper proposes Bayesian pseudocoresets to improve the computational complexity of Exemplar-VAE and to reduce overfitting. The paper is well-written, and extensive experimental evaluation is provided. Variational inference to find optimal pseudocoresets, and the joint optimization along with VAE are strong technical contributions. However, the paper very closely follows the structure, formatting, and experimental evaluation of Exemplar-VAE (a related work). New experimental evaluation or insights on the pseudocoresets are not provided but a repetition of the same experiments conducted by Exemplar-VAE on the proposed approach. The novelty of the paper is thus somewhat limited and incremental. The pseudocode for the algorithm optimization is not explained in the text.*
>
> **R.** Thank you for the approval of our paper. Our model is developed for reducing the computational complexity of Exemplar-VAE. Thus, to better understand and distinguish the differences between two algorithms, we employ the same structure and formatting of Exemplar-VAE. And to further show the effectiveness of our method and the fairness of comparison, we also follow the experimental evaluation of Exemplar-VAE.
>
> To illustrate the benefits of introducing pseudocoreset, we additional measure the running time on the ﬁrst network architecture for three datasets (as shown in Table 2) and report the log-likelihood values of two models for the test set, as shown in Fig. 5. The experimental results show that our method can improve the efﬁciency.
>
> The derivations for the algorithm optimization of the pseudocode are provided in Supp. B and C. And more details on the optimization process would be obtained in the revised manuscript.
>
>
>
> **Q2.** *The optimization process in Algorithm 2 is not explained in the paper. What is the significance of subsets S and M in Algorithm 2?*
>
> **R.**  Thanks for pointing this out. We employ a two-step alternative optimization strategy for optimization. A more detailed explanation for the optimization process will be added in the revised manuscript. Subsets S denotes the samples ﻿from pseudocoreset posterior and M denotes the number of pseudodata points as shown in the line of 110.
>
>
>
> **Q3.** *Equations (9)-(14) are similar or same as the ones in Exemplar-VAE. The equations defined in Exemplar-VAE should be cited.*
>
> **R.** Since Exemplar-VAE and our method are variants of VAE and both formulate the algorithm as a variational inference problem, the structure of equations seems to be the same. In particular, we follow the formatting of Exemplar-VAE which makes these equations look more similar. However, the specific meanings of these equations are different. Equation (9) denotes the objective function of our method based on the pseudocoresets. For the definition of variational posterior shown on equation (10), we use the Gaussian distributions. This is consistent with not only Exemplar-VAE but also other VAE variants. Equation (11) denotes the prior conditioned on a pseudodata point. Moreover, equations (12-14) are come up through derivation. We will add the cite for the equations defined in Exemplar-VAE.
>
>
>
> **Q4.** *Statistical significance of results on KNN is not provided. The results on density estimation show marginal improvement for the proposed approach.*
>
> **R.**  Thanks for pointing this out.  We also record the statistical significance of kNN classification accuracy with different values of K in Fig.4 and Supp.F.  Due to the template, we can't display figures here, so we use the following Table.8 instead for as an example. In the revised version, we will modify Fig. 4 and Supp.F to better display.
>
> | K    | Gaussian       | Exemplar       | Vamprior       | BPE-VAE        |
> | ---- | -------------- | -------------- | -------------- | -------------- |
> | 3    | 82.02 +/- 0.06 | 82.69 +/- 0.13 | 82.91 +/- 0.25 | 83.19 +/- 0.03 |
> | 5    | 82.81 +/- 0.22 | 83.59 +/- 0.22 | 83.74 +/- 0.06 | 84.09 +/- 0.09 |
> | 7    | 83.13 +/- 0.31 | 84.05 +/- 0.11 | 84.23 +/- 0.21 | 84.42 +/- 0.20 |
> | 9    | 83.29 +/- 0.18 | 84.22 +/- 0.04 | 84.21 +/- 0.21 | 84.60 +/- 0.10 |
> | 11   | 83.21 +/- 0.14 | 84.38 +/- 0.12 | 84.34 +/- 0.14 | 84.77 +/- 0.12 |
> | 13   | 83.28 +/- 0.11 | 84.44 +/- 0.19 | 84.40 +/- 0.10 | 84.83 +/- 0.07 |
> | 15   | 83.30 +/- 0.25 | 84.37 +/- 0.10 | 84.17 +/- 0.22 | 84.83 +/- 0.08 |
>
> Table 8. kNN classiﬁcation accuracy (%) with different values of K on Fahsion MNIST
>
>
>
> **Q5.** *The t-SNE plots should also compare the latent space of Exemplar-VAE. Currently only the proposed approach and a baseline VAE are compared.*
>
> **R.** Thanks for pointing this out.  We report the t-SNE visualization of latent representations of MNIST on both four methods, including VAE with a Gaussian prior, VAE with a VampPrior, Examplar-VAE, and ByPE-VAE.  Because the figure cannot be displayed here, the results can be seen in the revised manuscript.
>
>
>
> **Q6.** *Experiments to show how well the learned pseudocoresets approximate the training data, would be a useful addition.*
>
> **R.** Using the pseudocoresets approximate the training data directly affects the variational posterior and the second term of the objective function, so we can report the value of the second term to measure the learned pseudocoresets. Specifically, we compared the KL loss of VampPrior and ByPE prior, which are both pseudo-input methods, and we can see that our method can get a better approximation from below Table 6. The details will be added to the revised manuscript.
>
> | Result             | Dynamic MNIST      | Fashion MNIST     | CIFAR10            |
> | ------------------ | ------------------ | ----------------- | ------------------ |
> | VAE w/ VampPrior   | 12.08 +/- 0.05     | 8.05 +/- 0.05     | 21.80 +/- 0.07     |
> | ByPE VAE           | **11.93 +/- 0.12** | **8.03 +/- 0.01** | **21.55 +/- 0.02** |
> | HVAE w/ VampPrior  | 12.20 +/- 0.06     | 8.15 +/- 0.03     | 22.34 +/- 0.14     |
> | ByPE HVAE          | **12.18 +/- 0.06** | **8.10 +/- 0.04** | **22.16 +/- 0.06** |
> | ConvHVAE VampPrior | 12.66 +/- 0.06     | 8.54 +/- 0.05     | 24.42 +/- 0.24     |
> | ByPE ConvHVAE      | **12.50 +/- 0.06** | **8.49 +/- 0.05** | **24.15 +/- 0.28** |
>
> Table 6. The comparision of KL loss on different datasets
>
> **Q7.** *A comparison of KNN/log-likelihood for different coreset sizes would be useful. How does the proposed approach perform if the coreset size is set equal to the exemplar size in Exemplar-VAE?*
>
> **R.** Figure 5 shows the average negative log-likelihood of Exemplar VAE and ByPE-VAE on the test set with different sample sizes. The coreset aims to improve computational efficiency. If the coreset size is set equal to the exemplar size in Exemplar-VAE, the computational complexity will be greatly increased, which also means that the advantage of the coreset will be lost. At the same time, we can see from Figure 5 that when its size exceeds 500, the value has tended to converge.

---

> > ### Comment · Reviewer_rLyr · 2021-08-24
> > **Authors' reply**
> >
> > After reading the authors' reply, I am okay with a Marginally above the acceptance threshold

---

> > > ### Author Response · Authors · 2021-08-26
> > > **Thanks**
> > >
> > > Thank you again for approval of our work and your time spending.

---

### Official Review · Reviewer_X1mj · 2021-07-22

**Rating:** 6
**Confidence:** 3

**Summary:**

The paper tackles the important problem of finding suitable learnable prior for VAE. The paper provides some improvements of Exemplar VAE that increase the efficiency of the model. The proposed model is simply aiming at reducing the examples for nearest neighbours used by Exemplar VAE by application of bayesian pseudocoresets  for efficiently approximating the entire original dataset. The proposed approach was evaluated using standard benchmark datasets in a number of tasks, including density estimation, representation learning and generative data augmentation.

**Limitations And Societal Impact:**

I did not notice the limitations of proposed approach - it should be expressed in more visible way. The negative aspects are not discussed by authors - also needs including.

**Main Review:**

The paper is well-written and easy to follow. The main contribution of the paper is well-expressed and seems to be well-motivated. I like the application for data augmentation - however it requires more experimental analysis.

The proposed approach seems to be a bit incremental compared to Exemplar VAE  and VAMPrior. Conceptually, the paper presents the alternative way (referring to VamPrior) of learning pseudo inputs for the Exemplar VAE model. It would be beneficial to make some deeper comparison (both theoretical nad empirical) between the nature of pseudo inputs from VAMPrior and ByPE-VAE and express the novelty.

After reading the introduction and method description I would expect more convincing results of experiments. It is not clear, what type of criterion is provided in table 1. It seems that it is test NLL estimated using the Importance Sampling. Why pure ELBO value is not used for evolution? For CelebA there are no quantitative results reported in Table 1. but this dataset was used in qualitative analysis.

For the remaining experiments the range of datasets is reduced to MNIST and related. Having the models trained on CIFAR10 and CelebA it would be beneficial to see complete results for augmentation and representation tasks, even some of them are not SOTA compared to reference approaches. Similar issue is observed with table 2. - why results for VAMP (assuming the same number of pseudo inputs) are not reported here.

Because I serve as emergency reviewer I did not have to chance to deeply go through the math formulas but they seems to be correct.

**Time Spent Reviewing:**

2

---

> ### Author Response · Authors · 2021-08-09
> **Response to reviewer X1mj**
>
> # Rebuttal to Reviewer 1
>
> **Q1.** *The paper is well-written and easy to follow. The main contribution of the paper is well-expressed and seems to be well-motivated. I like the application for data augmentation - however it requires more experimental analysis.*
>
> **R.** Thank you for the approval of our paper and the insightful comments. For data augmentation, we report the performance of our method on MNIST in Table 3. To better evaluate the effectiveness of our methods, we add three additional sets of experiments.  First, we test the performance on Fashion MNIST and CIFAR10, respectively. Then, we report the performance of our method based on different values of $\lambda$​​​​. The results are summarized in the Tables(Table.4 and Table.5) below, which show that our method is mostly more effective than other VAEs for data augmentation.
>
> | Model                                   | Test-error on Fashion MNIST | Test-error on CIFAR10 |
> | --------------------------------------- | --------------------------- | --------------------- |
> | Gaussian prior w/ Variational Posterior | 9.98 +/- 0.08               | 50.07 +/- 0.23        |
> | Vampprior w/ Variational Posterior      | 10.03 +/- 0.05              | 50.74 +/- 0.16        |
> | Exemplar prior w/ Variational Posterior | **9.46** +/- 0.02           | 49.59 +/- 0.21        |
> | ByPE-VAE w/ Variational Posterior       | 9.75 +/- 0.02               | **49.00** +/- 0.04    |
> | Exemplar prior w/ Prior                 | 9.58 +/- 0.01               | 47.20 +/- 0.13        |
> | ByPE-VAE w/ Prior                       | **9.56** +/- 0.02           | **46.60** +/- 0.13    |
>
> Tabel 4. Test error (%) on permutation invariant Fashion MNIST and CIFAR10.
>
> | Lambda     | 0           | 0.1         | 0.2         | 0.3         | 0.4         | 0.5         | 0.6         | 0.7         | 0.8         | 0.9         | 1.0         |
> | ---------- | ----------- | ----------- | ----------- | ----------- | ----------- | ----------- | ----------- | ----------- | ----------- | ----------- | ----------- |
> | Test-error | 1.42+/-0.08 | 1.16+/-0.10 | 0.93+/-0.01 | 0.96+/-0.01 | 0.88+/-0.02 | 0.83+/-0.05 | 0.90+/-0.02 | 0.94+/-0.02 | 0.98+/-0.01 | 1.06+/-0.01 | 1.31+/-0.00 |
>
> Table 5. MNIST test error versus λ, which controls the relative balance of real and augmented data for ByPE-VAE
>
> **Q2.** *The proposed approach seems to be a bit incremental compared to Exemplar VAE and VAMPrior. Conceptually, the paper presents the alternative way (referring to VamPrior) of learning pseudo inputs for the Exemplar VAE model. It would be beneficial to make some deeper comparison (both theoretical nad empirical) between the nature of pseudo inputs from VAMPrior and ByPE-VAE and express the novelty.*
>
> **R.** While the purpose of using pseudo-inputs of ByPE-VAE is similar to that of VampPrior, there are fundamental differences in theory. The pseudo-inputs in VampPrior are regarded as hyperparameters of the prior, which are obtained through backpropagation along with the parameters of the model.  However,  the pseudo-inputs of our method are based on the paradigm of coresets, which aims to find a small weighted subset for approximating the whole dataset. In this case, the pseudo-inputs learned by our method could approximate all the original data, with the weighting operation carried on. Specifically, we employ a speciﬁc form of coresets, namely Bayesian pseudocoresets, which are obtained by minimizing the KL divergence between the prior based on the pseudocoreset and that based on the whole dataset.
>
> In terms of experiment, to measure these two different pseudo-inputs, we could compare the value of the KL divergence between the prior distribution and the variational posterior distribution. The results (shown in Table 6) show our method mostly outperforms VampPrior on three datasets, indicating that the pseudo-inputs learned by our method are better.
>
> Meanwhile, ByPE-VAE is an improvement to Exemplar VAE. So, while improving efficiency, ByPE-VAE retains the advantages of the diversity of data generation compared to VampPrior.
>
> | Result             | Dynamic MNIST      | Fashion MNIST     | CIFAR10            |
> | ------------------ | ------------------ | ----------------- | ------------------ |
> | VAE w/ VampPrior   | 12.08 +/- 0.05     | 8.05 +/- 0.05     | 21.80 +/- 0.07     |
> | ByPE VAE           | **11.93 +/- 0.12** | **8.03 +/- 0.01** | **21.55 +/- 0.02** |
> | HVAE w/ VampPrior  | 12.20 +/- 0.06     | 8.15 +/- 0.03     | 22.34 +/- 0.14     |
> | ByPE HVAE          | **12.18 +/- 0.06** | **8.10 +/- 0.04** | **22.16 +/- 0.06** |
> | ConvHVAE VampPrior | 12.66 +/- 0.06     | 8.54 +/- 0.05     | 24.42 +/- 0.24     |
> | ByPE ConvHVAE      | **12.50 +/- 0.06** | **8.49 +/- 0.05** | **24.15 +/- 0.28** |
>
> Table 6. The comparision of KL loss on different datasets
>
> **Q3.** *After reading the introduction and method description I would expect more convincing results of experiments. It is not clear, what type of criterion is provided in table 1. It seems that it is test NLL estimated using the Importance Sampling. Why pure ELBO value is not used for evolution?*
>
> **R.** In the testing phase, the log-likelihood is desired to report for density estimation. Importance Sampling $^{[1]}$ is a direct way to estimate log-likelihood, which is a commonly used evaluation criterion in density estimation. However, the ELBO is just the lower bound of log-likelihood. Thus, we use the estimated NLL via Importance Sampling as the criterion. This measurement is also consistent with that of  Exemplar VAE and VampPrior.
>
> [1] Burda Y, Grosse R, Salakhutdinov R. Importance weighted autoencoders[J]. arXiv preprint arXiv:1509.00519, 2015.
>
> **Q4.** *For CelebA there are no quantitative results reported in Table 1. but this dataset was used in qualitative analysis.*
>
> **R.** Thanks for providing thoughtful feedback. We also report the test negative log-likelihood(NLL) for CelebA, as shown in the Table.7 below. The experimental results show that ByPE-VAE outperforms other models.
>
> | Method         | Test-loglikelihood      |
> | -------------- | ----------------------- |
> | Gaussian prior | 183.16 +/- 0.40         |
> | Vampprior      | 183.61+/- 0.69          |
> | Exemplar       | 185.03 +/- 1.46         |
> | ByPE           | **182.11** **+/- 1.10** |
>
> Tabel 7. Density estimation on CelebA based on the Fully Convolutional Neural Network.
>
> **Q5.** *For the remaining experiments the range of datasets is reduced to MNIST and related. Having the models trained on CIFAR10 and CelebA it would be beneficial to see complete results for augmentation and representation tasks, even some of them are not SOTA compared to reference approaches. Similar issue is observed with table 2. - why results for VAMP (assuming the same number of pseudo inputs) are not reported here.*
>
> **R.** Thanks for your valuable comments. We show more experimental results on Fashion MNIST and  CIFAR10 for representation and data augmentation tasks in Table 4.  Consistent with Exemplar VAE, CelebA is mainly used to capture the generation ability in our experiments, and it is rarely applied to the classification task.  We also add the results of VampPrior in Table 2.
>
>
> | **Dataset**   | **Bype VAE(500)** | **Vamp VAE(500)** | **Exemplar VAE(25000)** |
> | ------------- | ----------------- | ----------------- | ----------------------- |
> | Dynamic MNIST | 23.61 / 13.19     | 23.65 / 13.03     | 23.61 / 35.45           |
> | Fashion MNIST | 20.85 / 12.05     | 20.87 / 12.09     | 20.81 / 37.23           |
> | CIFAR10       | 71.91 / 17.30     | 71.97 / 14.89     | 72.00 / 66.85           |
>
> Table 2. Result of average negative log-likelihood and training time(s/eopch) with various datasets

---

> > ### Comment · Reviewer_X1mj · 2021-08-20
> > **Thanks**
> >
> > Dear authors, thank you for the detailed comments. After reading the paper once again, reading the remaining reviews I can now admire more the contribution with respect to Vamp and Exemplar VAE. It was also impressive, that you managed to prepare large number of additional experiments in a very short time. However, the large number of additional experiments means that the base manuscript requires major revision, therefore, I upgrade my score only to 6/10.

---

> > > ### Author Response · Authors · 2021-08-26
> > > **Thanks**
> > >
> > > Thank you again for your approval of our work and insightful comments. We will revise our work in detail in the revised edition, especially the experimental section.

---

### Decision · Program_Chairs · 2021-09-27

**Decision:**

Accept (Poster)

**Comment:**

The paper proposes a new way of learning a prior in the Variational Auto-Encoder framework. The paper is a follow-up on the VampPrior VAE and the Exemplar VAE. In a nutshell, the idea is to use a Bayesian pseudocoreset to learn pseudoinputs (in the context of the VampPrior). Overall, the reviewers appreciate the paper because:
- It is easy to follow.
- All concepts are clearly explained.
- The idea of using the Byesian pseudocoreset is interesting.

The main concern is about the experimental section. The reviewers would prefer to see a deepr comparison of learned pseudoinputs using backpropagation (like in the VampPrior) or by the proposed approach, and how learned pseudoinputs compared to real data.
The rebuttal contains many new results and addresses many concerns raised by the reviewers. Therefore, I tend to accept the paper.